# Generalization and Learnability in Multiple Instance Regression

Kushal Chauhan[1]    Rishi Saket[1]    Lorne Applebaum[2]    Ashwinkumar Badanidiyuru[2]    Chandan Giri[2]

Aravindan Raghuveer[1]

[1]Google Research, India , {kushalchauhan,rishisaket,araghuveer}@google.com
[2] Google, USA , {lapplebaum,ashwinkumarbv,chandangiri}@google.com

## Abstract

Multiple instance regression (MIR) was introduced by Ray and Page [2001] as an analogue of multiple instance learning (MIL) in which we are given bags of feature-vectors (instances) and for each bag there is a bag-label which matches the label of one (unknown) primary instance from that bag. The goal is to compute a hypothesis regressor consistent with the underlying instance-labels. A natural approach is to find the best primary instance assignment and regressor optimizing the mse loss on the bags though no formal generalization guarantees were known. Our work is the first to prove generalization error bounds for MIR when the bags are drawn i.i.d. at random. Essentially, with high probability any MIR regressor with low error on sampled bags also has low error on the underlying instance-label distribution. We next study the complexity of linear regression on MIR bags, shown to be NP-hard in general by Ray and Page [2001], who however left open the possibility of arbitrarily good approximations. Significantly strengthening previous work, we prove a strong inapproximability bound: even if there exists zero bag-loss MIR linear regressor on a collection of 2-sized bags with labels in $[-1, 1]$, it is NP-hard to find an MIR linear regressor with bag-loss $< C$ for some absolute constant $C > 0$. Our work also proposes a model training method for MIR based on a novel weighted assignment loss, geared towards handling overlapping bags which have not received much attention previously. We conduct empirical evaluations on synthetic and real-world datasets showing that our method outperforms the baseline MIR methods.

## 1  INTRODUCTION

In traditional supervised learning, the training data consists of labeled instances represented by feature-vectors. In many applications however, due lack of instrumentation, uncertainty in the data or privacy constraints, instance-wise labels may not be available. Instead, the data consists sets or *bags* of instances and one label per such bag which is thought to depend on the (unknown) instance-labels present in the bag via some label aggregation function.

The approach of *multiple instance learning* (MIL) – introduced in [Dietterich et al., 1997] for predicting drug activity – trains an instance-level predictor to be consistent with the bag-labels of the training data according to the aggregation function. In the many commonly studied binary $\{0, 1\}$-label scenarios [Dietterich et al., 1997, Maron and Lozano-Pérez, 1997, Zhang and Goldman, 2001, Chen et al., 2006], the bag-label is OR i.e, disjunction of the instance-labels in the bag.

Our focus is *multiple instance regression* (MIR) introduced by [Ray and Page, 2001] as an analogue of MIL, in which the labels are real-valued and only one *primary* instance in a bag determines the bag label. These primary instances are unknown and the task here is to learn an instance-label predictor so that a primary instance per bag can be identified whose predicted label is consistent with the bag label. The trained instance-label predictor is then deployed to infer the label of unlabelled instances encountered in the future. The model training is formulated as an optimization problem: find an instance-level predictor and identify the primary instance of each training bag whose predicted label is taken to be the predicted bag-label. The objective is to minimize the loss between the observed and predicted bag-labels, where the typical loss metric is mean-squared error (mse). Given a predictor, the optimal primary instance for any bag is the one whose predicted label minimizes loss. The MIR formulation has been used to model applications in remote sensing such as aerosol optical depth prediction [Wang et al., 2008] and crop yield prediction [Wagstaff and Lane, 2007].

More recent works have applied MIR across multiple areas. In a novel deployment of MIR, the work of Serafini et al. [2022] used it to model electrical load disaggregation. In the biological domain, Park et al. [2020] have used MIR to model the continuous response of bags of neoantigens. For image quality assessment where each image patch has a probability of being prime, the work of Liang et al. [2021] applied an MIR approach to train a CNN. A different image analysis task - estimating facial age from images - has also been tackled using MIR techniques [Liu et al., 2019]. Previous works have proposed baselines which preprocess the data to produce fully supervised training data [Wang et al., 2008]; along with specialized methods based on expectation-maximization (EM) [Ray and Page, 2001, Wang et al., 2012] as well as clustering [Trabelsi and Frigui, 2018]. Most of the previous works consider the restricted case of disjoint bags, however overlapping bags occur in real-world applications such as electrical load disaggregation across time [Serafini et al., 2022] mentioned above, as well as continuous time human emotion recognition [Romeo et al., 2022] and musical clip analysis for automatic metadata tagging [Mandel and Ellis, 2008], the latter two applications, while studied more from the classification standpoint, admit analogous regression tasks.

While there has been substantial work on the applied aspects of MIR, a formal treatment from the statistical and computational perspectives has been lacking. Another aspect that has received little attention is the case of overlapping bags. For bags of size 1 i.e., traditional supervised regression, generalization error bounds are known depending on the complexity of the regressor class. Moreover, supervised linear regression (under mse) is known to be tractable on any distribution, in particular finding a perfect linear regressor i.e., with zero loss, as long as one exists, is computationally easy. To the best of our knowledge, these aspects for MIR have not (or only partially) been studied, for e.g. under what conditions will an instance-label predictor trained on a sample of MIR bags generalize to the underlying instance distribution? On the complexity side, while [Ray and Page, 2001] showed that computing the optimum mse-loss linear regression MIR with one primary instance per bag is NP-hard in general, they leave open the possibility of arbitrarily close approximations to the optimum.

Our work is the first to rigorously address the above questions. We first prove a bag to instance generalization error bound when bags are sampled i.i.d. at random, essentially showing that a regressor with bounded *pseudo-dimension* (see Sec. 2.2) with values in $[0, O(1)]$, where $O(1)$ denotes a constant, optimizing MIR on such bags also generalizes well on the underlying instance distribution. We informally state our result below:

**Theorem 1.1** (bag to instance generalization bound, informal). *Let $f^* : \mathcal{X} \to [0, O(1)]$ for some domain of real feature-vectors $\mathcal{X}$. Suppose $m$ i.i.d. MIR bags are sampled,* *each bag $B$ consisting of $k$ i.i.d. random instances from some distribution $\mathcal{D}$ over $\mathcal{X}$, with bag label $f^*(\mathbf{x})$ for a uniformly sampled $\mathbf{x} \in B$. Let $\mathcal{F}$ be a concept class of regressors which map $\mathcal{X}$ to $[0, O(1)]$, and for any $f \in \mathcal{F}$, $\varepsilon_{\text{MIR}}$ be its mse-loss on the sampled bags, while $\varepsilon_{\mathcal{D}}$ be its instance-wise mse-loss under $\mathcal{D}$. Then, with probability $1 - \delta$, $\varepsilon_{\mathcal{D}} \leq O\left(\varepsilon_{\text{MIR}}^{1/(2k+1)}\right)$ as long as $m/(\log m) > O\left((d/\varepsilon_{\text{MIR}}^{2k/(2k+1)})[\log(1/\delta) + \log(1/\varepsilon_{\text{MIR}})]\right)$ where $d$ is the pseudo-dimension of $\mathcal{F}$.*

Next we consider the problem of optimizing the loss of linear regression for MIR and show that it is NP-hard to approximate, even if a perfect solution exists and all bags are of size $\leq 2$.

**Theorem 1.2** (inapproximability of bag loss, informal). *Given an instance of MIR whose bags are of size $\leq 2$ with labels in $[-1, 1]$ such that there exists a linear regressor and primary instances for each bag whose label given by the linear regressor equals the bag label, it is NP-hard to find a linear regressor with primary instances per bag such that the optimum is strictly less than some absolute constant $c_0 > 0$ with respect to the mse-loss.*

From a more practical standpoint as well, our work focuses on the case of overlapping bags. These arise in applications in which instances can belong to several groups based either on temporal characteristics or annotations (see for e.g. [Serafini et al., 2022], [Romeo et al., 2022] and [Mandel and Ellis, 2008]). Such overlapping bag setting can also be constrained to ensure that an instance is primary for at most one bag that contains it. This *injectiveness* constraint is superfluous in the disjoint bag setting considered in most previous MIR methods, and therefore some of those techniques – such as assigning the bag label to all instances in that bag, or predicting the likelihood of an instance being primary independent of the bag – are either not applicable or don't result in a solution respecting that constraint.

We propose the *Weighted Assignment* model training that applies to overlapping bags along with injectiveness constraints. The method trains a label predictor model along with free trainable variables (one for each bag and instance in that bag) which model the primary instances in different bags. These variables are constrained via regularization terms to approximately be $\{0, 1\}$-valued, and sum to 1 within a bag, and are used to minimize loss between the prediction for the bag and its bag-label. Another regularization term across bags is used to make sure that an instance is primary for at most one bag.

We believe that MIR is of current practical relevance and our theoretical insights can impact the design and analysis of new techniques for MIR and are therefore interesting. The hardness result in Theorem 1.2 shows that this problem becomes hard in a strong sense when we have bags of size

2, and rules out a straightforward application of the simple techniques that can solve the instance-wise (bag size 1) case. Our generalization error bound (Theorem 1.1) is the first such for the MIR problem - it shows that optimizing the bag-level mse loss provably (in the case of random bags) learns the underlying instance labeling. This justifies our algorithmic approach explained above, the Weighted Assignment model training, for finding the prime instance assignment and the regressor to optimize the bag-level loss.

## 1.1 PREVIOUS RELATED WORK

The work of [Dietterich et al., 1997] introduced the classification setting of multiple instance learning (MIL) in the context of drug activity detection where the bag label is modeled as an OR of the its (unknown) instance-labels (all labels are $\{0, 1\}$-valued). Given a such a dataset of bags the goal is to train an instance-label predictor. This formulation was shown thereafter to have applicability in several other domains including the analysis of medical images [Wu et al., 2015] and videos [Sikka et al., 2013], information retrieval [Lozano-Pérez and Yang, 2000], time series prediction [Maron, 1998] and drug discovery [Maron and Lozano-Pérez, 1997]. Multiple instance regression (MIR) [Ray and Page, 2001] is the regression analogue in which the labels are real-valued and a *primary* instance in a bag determines the bag label. Other related settings in which aggregation occurs are *learning from label proportions* Rueping [2010], Wojtusiak et al. [2011] in which a bag's label is the average labels of its instances, and *distribution regression* Poczos et al. [2013], Szabó et al. [2016] where the bag denotes a probability distribution which is typically represented by a collection of samples from it.

Techniques such as maximum-likelihood or boosting using differentiable approximations to the OR function [Ramon and De Raedt, 2000, Zhang et al., 2005] and logistic regression [Ray and Craven, 2005] were proposed. More specialized MIL techniques include the diverse-density (DD) method [Maron and Lozano-Pérez, 1997] and and its EM-based variant, EM-DD [Zhang and Goldman, 2001]. On the theoretical front, Blum and Kalai [1998] showed that noise tolerant PAC learnability implies MIL PAC learnability for i.i.d. bags while Sabato and Tishby [2012] showed generalization bounds for the classification error on bags.

While MIL in the classification setting has been extensively studied, the MIR problem has received much less attention, its study largely being specific to the remote sensing domain. Straightforward baseline methods transform the problem into a fully supervised setting by either (i) averaging the feature-vectors in each bag and assigning it the bag label i.e., aggregated-MIR [Wang et al., 2008], or (ii) instance-MIR in which the bag-label is assigned to each instance in a bag (see [Ray and Craven, 2005]). More sophisticated EM based methods were developed, first of which was primary-MIR

(PIR) [Ray and Page, 2001] followed by others such as pruning MIR [Wang et al., 2008] and mixture-model MIR [Wang et al., 2012], while other work [Wagstaff et al., 2008, Trabelsi and Frigui, 2018] proposed methods for MIR based on clustering techniques. Among these, aggregated-MIR and the pruning-MIR methods are applicable to overlapping bags as they operate at a bag level (collapsing or shrinking them).

## 1.2 OVERVIEW OF PROOF TECHNIQUES

**Bag to instance generalization error bound.** With the setup in the statement of Theorem 1.1, we prove the contrapositive with high probability: if there is a lower bound of $4\zeta$ on the instance-level error of any $f \in \mathcal{F}$, then for any prime instance assignment to bags, the loss on the sampled bags is at least $\Omega(\zeta^{2k+1})$ as long as the lower bound on $m$ in the statement holds. We can think of the $m$ i.i.d. bags being constructed as follows: sample $mk$ i.i.d. instances from $\mathcal{D}$ and then randomly partition them into $m$ bags of $k$ feature-vectors each, and from each bag select a primary instance at random and assign its label given by $f^*$ to the bag. The known generalization error bounds for regression imply that when $m$ satisfies the given lower bound, for any $f \in \mathcal{C}$ with loss on $\mathcal{D}$ at least $4\zeta$, its loss on the $mk$ sampled points is at least $2\zeta$. By losing another additive $\zeta$ in the loss we can restrict ourselves to such $f$ belonging to an appropriately fine-grained $\ell_\infty$-cover for $\mathcal{F}$. Since, the range of $f$ is bounded, we obtain by averaging that there must be a $\Omega(\zeta mk)$ points where the $f$ has regression loss at least $\zeta/2$.

In comparison to the fully supervised case, having bags of size $> 1$ affords more choice to a bad regressor $f$ - it can fit the bag-label by a low error prediction on any one of the instances in the bag. To show this is not possible with high probability for all bags, we show - using a bucketing argument - in the key Lemma 3.3 that among the $mk$ points sampled, there is a sizable subset $\mathcal{S}$ such that all the values of $f$ on $\mathcal{S}$ are far from all the values of $f^*$ on $\mathcal{S}$ (†), where $f^*$ is the instance-labeling. More formally, by a counting argument over a division of the range into $\zeta/4$-length segments, we obtain that for a subset $\mathcal{S}$ of size at least $\Omega(\zeta^2 mk/R^2) =: 2pmk$ of the sampled points, the value of $f^*$ on any of those points is at least $\zeta/4$ in distance from the value of $f$ on any of those points.

Lemma 3.4, via a combinatorial analysis of the sampling induced by the random partitioning, yields that with high probability at least $p^k$ fraction of the bags are sampled entirely from $\mathcal{S}$ each of which induces a loss of at least $\zeta/16$. In other words, a significant fraction of the sampled bags are subsets of $\mathcal{S}$. These bags induce the lower bound on the bag-level loss since $f$ is bound to incur a high error on these bags due to the property (†) above of $\mathcal{S}$. A further union bound on the $\ell_\infty$-cover of $\mathcal{C}$ yields the desired bound. Section 3 states our generalization error bound (Theorem

3.1) and includes its detailed proof.

**Hardness of approximating linear MIR.** The hardness reduction follows the (by now commonly used) template of combining a tailored *dictatorship test* with a hard to approximate constraint satisfaction problem (CSP). The dictatorship test – usually the key ingredient – is a toy version of the problem defined over some domain e.g. $\mathbb{R}^K$ which admits a good solution corresponding to each coordinate in $[K]$ (completeness), while on the other hand any good solution to the problem must depend significantly on at least one distinguished coordinate in $[K]$ (soundness). For our problem, we construct it as follows: let $\boldsymbol{\mathcal{X}} = \{-1, 1\}^K$, and for each $\mathbf{x} \in \boldsymbol{\mathcal{X}}$, add *two copies* of the bag $\{\mathbf{x}, -\mathbf{x}\}$, one with bag-label $-1$ and another with bag-label $1$, which can have different primary instances. For the completeness property, observe that for any $i \in [K]$ the regressor given by $f(\mathbf{x}) = x_i$ assigns $-1$ and $1$ to the two instances of each bag. Thus, by appropriately choosing the primary instances, their labels can match the corresponding bag-labels leading to a zero-loss solution.

For soundness, let us for ease of exposition restrict to only homogeneous linear regressors of the form $\langle \mathbf{r}, \mathbf{x} \rangle$ for some $\mathbf{r} \in \mathbb{R}^K$. Suppose that for all $i \in [K]$, $c_i \ll \|\mathbf{c}\|_2$ i.e., the regressor does not have any distinguished coordinate. Then, using Berry-Esseen theorem one can show that under the uniform distribution over $\boldsymbol{\mathcal{X}}$, $\langle \mathbf{r}, \mathbf{x} \rangle$ is distributed close to a mean-zero Gaussian. Now, it is easy to see that a random point from such a Gaussian and its negation, both are at a constant distance from the value $1$ with significant probability. This immediately yields a constant lower bound on the loss, demonstrating the soundness. This dictatorship test is plugged into a hard-to-approximate Label Cover problem with certain structural properties which, along with the technique of *folding* over the constraints, aid in the reduction's analysis which we omit in this overview. As evident in this discussion, our reduction creates overlapping bags. Nevertheless, a straightforward scaling perturbation can ensure that all bags are pairwise disjoint. In particular, our hardness result also applies to injective MIR.

Section 4 formally states our hardness result and includes the formal description and analysis of the dictatorship test. The rest of the proof is included in Appendix A along with an explanation in Appendix A.5 of the perturbation used to make the bags pairwise disjoint.

## 2 PRELIMINARIES

### 2.1 NOTATIONS AND PROBLEM DEFINITION

Let $\boldsymbol{\mathcal{X}}$ be a set of real feature-vectors i.e., $\boldsymbol{\mathcal{X}} \subseteq \mathbb{R}^{d_0}$ for some $d_0 \in \mathbb{Z}^+$. A *bag* $B$ is a subset of $\boldsymbol{\mathcal{X}}$. An instance $\mathcal{I}$ of MIR consists of a collection $\mathcal{B}$ of $m$ bags $\{B_1, \ldots, B_m\}$ along with a label vector $\boldsymbol{\sigma} = (\sigma_1, \ldots, \sigma_m)$ with the goal

being to find:

- a predictor $h : \boldsymbol{\mathcal{X}} \to \mathbb{R}$, and
- an assignment $\Gamma : \mathcal{B} \to \boldsymbol{\mathcal{X}}$ s.t. $\Gamma(B) \in B$ $\forall B \in \mathcal{B}$ indicating the *primary* instance i.e., feature-vector for each bag

minimizing the following objective

$$\mathsf{val}\left(L_{\mathrm{reg}}, \mathcal{I}, h, \Gamma\right) := \mathbb{E}_{j \in [m]}\left[L_{\mathrm{reg}}\left(\sigma_j, h\left(\Gamma(B_j)\right)\right)\right] \quad (1)$$

for some loss function $L_{\mathrm{reg}}$. For convenience we subsume the optimization over $\Gamma$ by defining:

$$\mathsf{val}\left(L_{\mathrm{reg}}, \mathcal{I}, h\right) := \min_{\Gamma} \mathbb{E}_{j \in [m]}\left[L_{\mathrm{reg}}\left(\sigma_j, h\left(\Gamma(B_j)\right)\right)\right] \quad (2)$$

If there are no other constraints on $\Gamma$, then clearly $\mathsf{val}\left(L, \mathcal{I}, h\right)$ is minimized when $\Gamma(B_j) = \arg\min_{\mathbf{x} \in B} L_{\mathrm{reg}}(\sigma_j, h(\mathbf{x}))$, $j \in [m]$. In the case the bags are overlapping one may add the constraints that for each $\mathbf{x}$, $|\{B \in \mathcal{B} \mid \mathbf{x} \in B, \Gamma(B) = \mathbf{x}\}| \leq 1$ i.e., an instance may be primary for at most one bag. We shall refer to this problem as *injective* MIR.

For brevity we shall denote by $\mathsf{val}_p\left(\mathcal{I}, h\right)$ the LHS of (2) when $L_{\mathrm{reg}}(a, b) := |a - b|^p$, for any $p \geq 1$. In particular, $\mathsf{val}_2$ uses the mse-loss.

Let $\mathcal{D}$ be a distribution on $\boldsymbol{\mathcal{X}}$. For some $f : \boldsymbol{\mathcal{X}} \to [0, R]$ and $k \in \mathbb{Z}^+$, an instance of IID-MIR$[f, k, m]$ is a random problem instance $\mathcal{I}$ of MIR with $m$ bags where independently for each $j \in [m]$:
(i) bag $B_j = \{\mathbf{x}_{1j}, \ldots, \mathbf{x}_{kj}\}$, where $\mathbf{x}_{ij} \sim \mathcal{D}$, independently for $i = 1, \ldots, k$, and
(ii) $\sigma_j = f(\mathbf{x}_{1j})$.

### 2.2 USEFUL CONCEPTS AND TOOLS

For our generalization error bound, we shall restrict ourselves to a class $\mathcal{F}$ of real-valued functions (regressors) over $\boldsymbol{\mathcal{X}}$ with values i.e., predictions in $[0, R]$ for some $R \in \mathbb{R}$ s.t. $R \geq 1$. For any $\boldsymbol{\mathcal{X}}' \subseteq \boldsymbol{\mathcal{X}}$ s.t. $|\boldsymbol{\mathcal{X}}'| = N$, let $\mathcal{C}_p(\xi, \mathcal{F}, \boldsymbol{\mathcal{X}}')$ denote a minimum cardinality $\ell_p$-metric $\xi$-cover of $\mathcal{F}$ over $\boldsymbol{\mathcal{X}}'$, for some $\xi > 0$. Specifically, $\mathcal{C}_p(\xi, \mathcal{F}, \boldsymbol{\mathcal{X}}')$ is a minimum sized subset of $\mathcal{F}$ such that for each $f^* \in \mathcal{F}$, there exists $f \in \mathcal{C}_p(\xi, \mathcal{F}, \boldsymbol{\mathcal{X}}')$ s.t. $\left(\mathbb{E}_{\mathbf{x} \in \boldsymbol{\mathcal{X}}'}\left[|f^*(\mathbf{x}) - f(\mathbf{x})|^p\right]\right)^{1/p} \leq \xi$ for $p \in [1, \infty)$, and $\max_{\mathbf{x} \in \boldsymbol{\mathcal{X}}'} |f^*(\mathbf{x}) - f(\mathbf{x})| \leq \xi$ for $p = \infty$.

The maximum size of such a cover over all choices of $\boldsymbol{\mathcal{X}}' \subseteq \boldsymbol{\mathcal{X}}$ s.t. $|\boldsymbol{\mathcal{X}}'| = N$ is defined to be $N_p(\xi, \mathcal{F}, N)$. In other words, such a cover of size $N_p(\xi, \mathcal{F}, N)$ always exists for $p = [1, \infty]$. We refer the reader to Sections 10.2-10.4 of [Anthony and Bartlett, 2009] for more details (see also Chapter 2.2 of Vaart and Wellner [1996]).

The *pseudo-dimension* of $\mathcal{F}$, $\mathsf{Pdim}(\mathcal{F})$ is a measure of the complexity of the of $\mathcal{F}$. As described in Sec. 10.4 and 12.3

of [Anthony and Bartlett, 2009], the pseudo-dimension can be used to bound the size of covers for $\mathcal{F}$ as follows:

$$N_1(\xi, \mathcal{F}, N) \leq N_\infty(\xi, \mathcal{F}, N) \leq (eNR/\xi d)^d \quad (3)$$

where $d = \mathsf{Pdim}(\mathcal{F})$ and $N \geq d$.

# 3 GENERALIZATION ERROR BOUND

With the setup in Sec. 2, let $f^* : \mathcal{X} \to [0, R]$ be any labeling. This section is devoted to proving the following theorem.

**Theorem 3.1.** *There is a constant $K_0 > 0$ s.t. for parameters $\varepsilon \in [0, R^2]$ and $\delta \in (0, 1)$, if $m/(\log m) \geq d \left(\frac{K_0 R^4}{\varepsilon^2}\right)^{2k} \left(\log\left(\frac{1}{\delta}\right) + \log\left(\frac{Rk}{\varepsilon}\right)\right)$, then with probability $1 - \delta$ over instance $\mathcal{I}$ of IID-MIR$[f^*, k, m]$: any $h \in \mathcal{F}$ s.t. $\mathbb{E}_\mathcal{D}\left[|h(\mathbf{x}) - f^*(\mathbf{x})|^2\right] \geq \varepsilon$ satisfies $\mathsf{val}_2(\mathcal{I}, h) \geq \varepsilon^{2k+1}/16R^{8k+1}$.*

The converse of the above theorem is obtained by letting $\varepsilon_{\text{MIR}} = \varepsilon^{2k+1}/16R^{8k+1}$ (along with some simplifications) and is stated below.

**Corollary 3.2.** *For the lower bound on $m$ above, with probability at least $1 - \delta$, $\mathsf{val}_2(\mathcal{I}, h) \leq \varepsilon_{\text{MIR}}$ implies that $\mathbb{E}_\mathcal{D}\left[|h(\mathbf{x}) - f^*(\mathbf{x})|^2\right] \leq \left(16\varepsilon_{\text{MIR}}\right)^{1/(2k+1)} R^{(8k+1)/(2k+1)} \leq \left(16\varepsilon_{\text{MIR}}\right)^{1/(2k+1)} R^4$, since $R \geq 1$.*

Let $\zeta > 0$ be a small enough parameter to be fixed later. For convenience we shall prove the generalization error bounds for the $\ell_1$-loss (mae) and then translate them to $\ell_2^2$-loss (mse).

## 3.1 FIXING THE INSTANCES

The process of sampling a random instance $I$ of IID-MIR$[f^*, k, m]$ can be equivalently defined as follows:

- Sample a collection $\mathcal{Z}$ of $mk$ i.i.d. points from $\mathcal{D}$.
- Randomly partition $\mathcal{Z}$ into $k$-sized bags $B_1, \dots, B_m$
- For each $j \in [m]$ choose a random feature-vector from $B_j$ and let its label under $f^*$ be the bag-label of $B_j$.

In this subsection, we shall prove bounds after fixing the underlying instances $\mathcal{Z}$ sampled in the above process.

### 3.1.1 Bag error lower bound for fixed $f$

Let $f \in \mathcal{F}$ be s.t.

$$\mathbb{E}_{\mathbf{x} \in \mathcal{Z}}\left[|f^*(\mathbf{x}) - f(\mathbf{x})|\right] > \zeta \quad (4)$$

We have the following lemma.

**Lemma 3.3.** *There is $\mathcal{S} \subseteq \mathcal{Z}$ s.t. $|\mathcal{S}| \geq \zeta^2 mk/(10R^2)$ and for any $\mathbf{x}, \mathbf{z} \in \mathcal{S}$, $|f^*(\mathbf{x}) - f(\mathbf{z})| > \zeta/4$.*

*Proof.* Note that $\max_{\mathbf{x} \in \mathcal{X}} |f^*(\mathbf{x}) - f(\mathbf{x})| \leq R$. Using this upper bound along with (4) and an averaging argument we obtain that $\exists \mathcal{S}_0 \subseteq \mathcal{Z}$ s.t. $|\mathcal{S}_0| \geq \zeta mk/(2R)$ and for any $\mathbf{x} \in \mathcal{S}_0$, $|f^*(\mathbf{x}) - f(\mathbf{x})| > \zeta/2$, if not, then the LHS of (4) is at most $R\zeta/(2R) + (1 - \zeta/(2R))(\zeta/2) < \zeta$ which is a contradiction. For $i \in \{1, \dots, \lceil 4R/\zeta \rceil\}$ define $\mathcal{S}_i = \{\mathbf{x} \in \mathcal{S}_0 \mid f^*(\mathbf{x}) \in [(i-1)\zeta/4, i\zeta/4]\}$. Note that by the construction of $\mathcal{S}_0$, for each $i \in \{1, \dots, \lceil 4R/\zeta \rceil\}$ and any $\mathbf{x}, \mathbf{z} \in \mathcal{S}_i$, $|f^*(\mathbf{x}) - f(\mathbf{z})| > \zeta/4$. Choose $\mathcal{S}$ to be the $\mathcal{S}_i$ with the largest size – which is at least $mk(\zeta/(2R)) / (\lceil 4R/\zeta \rceil)$. Note that $\zeta \leq R$ and therefore $\lceil 4R/\zeta \rceil \leq 4R/\zeta + 1 \leq 4R/\zeta + R/\zeta \leq 5R/\zeta$. Thus, we obtain that $|\mathcal{S}| \geq \zeta^2 mk/(10R^2)$, completing the proof. $\square$

Define $\upsilon := \zeta^2/(20R^2)$ so that $\zeta^2 mk/(10R^2) = 2\upsilon mk$. Let $p := |\mathcal{S}|/(2|\mathcal{Z}|) = |\mathcal{S}|/(2mk) \geq \upsilon$. We now show that (in the random partitioning step), a significant number of bags have all the elements from $\mathcal{S}$.

**Lemma 3.4.** *With probability at least $1 - 2\exp\left(-m\upsilon^k/8\right)$ the number of bags having all $k$ elements from $\mathcal{S}$ is at least $m\upsilon^k/2$.*

*Proof.* After the randomized partitioning step each bag gets a certain number of elements from $\mathcal{S}$ and the rest from $\mathcal{Z} \setminus \mathcal{S}$, with exactly $k$ elements per bag. We model this process of generating these counts as follows:

1. Initially each of the $m$ bags have $k$ uncolored balls each, in total having $mk$ balls.
2. Each uncolored ball independently is colored *red* with probability $p$ and with probability $1 - p$ colored blue. Depending on the total number of red balls go to either Step 3 or Step 4.
3. If the total number red balls exceeds $|\mathcal{S}| = 2pmk$ by $r$, then a random choice of $r$ red balls are colored blue.
4. If the total number red balls is less than $|\mathcal{S}| = 2pmk$ by $r$, then a random choice of $r$ blue balls are colored red.

In the end, a random set of exactly $|\mathcal{S}|$ balls are colored red, and therefore the distribution of red-ball counts in the bags is same as that of the number of elements of $\mathcal{S}$ in the partitioning process. Thus, all we need to estimate is the number of bags with $k$ red balls. Since the ball coloring step in Step 2 is i.i.d. random, each bag independently gets $k$ red balls with probability $p^k$. Letting $s$ be the number of bags with $k$ red balls, by the lower tail Chernoff bound (see Theorem 4 in [Goemans, 2015]), we obtain at Step 2 that $\Pr[s \geq mp^k/2] \geq 1 - \exp\left(-mp^k/8\right)$. Now, if we are in Step 4 then the number of bags with all red balls does not decrease, while it Step 3 this number can decrease. So we only need to subtract off the probability that Step

3 happens, which by the Chernoff bound (upper tail) is at most $\exp\left(-mp^k/3\right) \leq \exp\left(-mp^k/8\right)$. Thus, at the end of the process, $\Pr[s \geq mp^k/2] \geq 1 - 2\exp\left(-mp^k/8\right)$ completing the proof. $\qquad\square$

### 3.1.2 Union bound over cover

As described in Sec. 2.2, let $\mathcal{C}_\infty(\xi, \mathcal{F}, \mathbf{\mathcal{Z}})$ be an $\ell_\infty$-metric $\xi$-cover whose size we shall denote for convenience by $q_\infty$, for some parameter $\xi > 0$ we shall set later. Then, from Lemma 3.4 and by union bound we obtain that with probability at least $1 - 2q_\infty\exp\left(-mv^k/8\right)$ the following event $E_0$ holds: for each $f \in \mathcal{C}_\infty(\xi, \mathcal{F}, \mathbf{\mathcal{Z}})$ satisfying (4)

- the number of bags having all $k$ elements from the corresponding $\mathbf{\mathcal{S}}$ (see Lemma 3.3) is at least $mp^k/2 \geq mv^k/2$. Call these the $\mathbf{\mathcal{S}}$-covered bags with respect to $f$.

### 3.2 ERROR BOUNDS FOR $\mathbf{\mathcal{Z}}$

Consider the subset $\mathcal{F}_{\text{err}} \subseteq \mathcal{F}$ of all $h \in \mathcal{F}$ such that $\mathbb{E}_\mathcal{D}\left[|h(\mathbf{x}) - f^*(\mathbf{x})|\right] \geq \hat\zeta$ for $\hat\zeta := 4\zeta$. We have the following lemma.

**Lemma 3.5.** *With probability at least* $1 - 4q_1\exp\left(-\hat\zeta^2 mk/(128R^2)\right)$ *over the choice of* $\mathbf{\mathcal{Z}}$,

$$\forall h \in \mathcal{F}_{\text{err}} \quad \mathbb{E}_{\mathbf{x} \in \mathbf{\mathcal{Z}}}\left[|h(\mathbf{x}) - f^*(\mathbf{x})|\right] \geq \hat\zeta/2 \quad (5)$$

*where* $q_1 = N_1(\hat\zeta/32, \mathcal{F}, 2mk)$.

*Proof.* The labeling is given by $f^*$ and therefore the true error of $h$ is $\mathbb{E}_\mathcal{D}\left[|h(\mathbf{x}) - f^*(\mathbf{x})|\right]$. The empirical error is $\mathbb{E}_{\mathbf{x} \in \mathbf{\mathcal{Z}}}\left[|h(\mathbf{x}) - f^*(\mathbf{x})|\right]$ where the expectation is over $\mathbf{x}$ sampled uniformly at random from $\mathbf{\mathcal{Z}}$. Therefore, given that $\mathbb{E}_\mathcal{D}\left[|h(\mathbf{x}) - f^*(\mathbf{x})|\right] \geq \hat\zeta$ for all $h$ in $\mathcal{F}_{\text{err}}$, the condition of (5) follows by an upper bound of $\hat\zeta/2$ on the difference between true and empirical errors for the class $\mathcal{F}_{\text{err}}$ given by Theorem 17.1 of [Anthony and Bartlett, 2009]. Since the mappings in the latter are to $[0,1]$ instead of $[0,R]$ in our case, we apply Theorem 17.1 of Anthony and Bartlett [2009] with $f^*/R$ as the labeling and $\overline{\mathcal{F}}_{\text{err}} := \{h/R : h \in \mathcal{F}_{\text{err}}\}$ as the function class. This amounts to taking $\hat\zeta/(2R)$ as the error $\varepsilon$ in Theorem 17.1 of Anthony and Bartlett [2009]. Observing that $N_1(\hat\zeta/(32R), \overline{\mathcal{F}}_{\text{err}}, 2mk) = N_1(\hat\zeta/(32), \mathcal{F}_{\text{err}}, 2mk)$ and that $|\mathbf{\mathcal{Z}}| = mk$ completes the argument. $\qquad\square$

Suppose the random choices of $\mathbf{\mathcal{Z}}$ and $\mathcal{B}$ ensure that (5) holds and letting $\xi := \zeta/8$ the event $E_0$ in the previous subsection also holds. This happens with probability $1 - 2q_\infty\exp\left(-mv^k/8\right) - 4q_1\exp\left(-\hat\zeta^2 mk/(128R^2)\right)$.

Consider any $h \in \mathcal{F}_{\text{err}}$ and let $f \in \mathcal{C}_\infty(\xi, \mathcal{F}, \mathbf{\mathcal{Z}})$ be the nearest to it in $\ell_\infty$-distance i.e, $\max_{\mathbf{x} \in \mathbf{\mathcal{Z}}} |h(\mathbf{x}) - f(\mathbf{x})| \leq \xi$. Noting that $\xi = \zeta/8$, from (5) and the triangle inequality, $f$ satisfies (4). Since the event $E_0$ holds, any $\mathbf{\mathcal{S}}$-covered bag with respect to $f$ incurs a bag-loss of at least $\zeta/4$ in $\text{val}_1(\mathcal{I}, h)$. By the nearness of $h$ and $f$ such a bag incurs a bag loss of at least $\zeta/4 - \xi \geq \zeta/8$ in $\text{val}_1(\mathcal{I}, h)$. By the lower bound on the number of such bags implied by $E_0$ we obtain $\text{val}_1(\mathcal{I}, h) \geq \zeta v^k/16$.

Summarizing the above we have that with probability at least:

$$1 - 2q_\infty\exp\left(-\frac{mv^k}{8}\right) - 4q_1\exp\left(-\frac{\zeta^2 mk}{8R^2}\right) \quad (6)$$

for all $h \in \mathcal{F}$ such that $\mathbb{E}_\mathcal{D}\left[|h(\mathbf{x}) - f^*(\mathbf{x})|\right] \geq 4\zeta$, $\text{val}_1(\mathcal{I}, h) \geq \zeta v^k/16$.

### 3.3 BOUNDS USING PSEUDO-DIMENSION

From Sec. 2.2 we have that,

$$q_\infty \leq \left(\frac{2emk}{d\xi}\right)^d, \quad q_1 \leq \left(\frac{8emk}{d\zeta}\right)^d \quad (7)$$

Using the above, there is some absolute constant $K_0 > 0$ s.t. choosing

$$\frac{m}{\log m} \geq d\left(\frac{K_0 R^2}{\zeta^2}\right)^{2k}\left(\log\left(\frac{1}{\delta}\right) + \log\left(\frac{k}{\zeta}\right)\right) \quad (8)$$

yields that (6) is at least $1 - \delta$, for $\delta \in (0, 1]$.

**MSE Error Bound.** Suppose that $h \in \mathcal{F}$ satisfied $\mathbb{E}_\mathcal{D}\left[|h(\mathbf{x}) - f^*(\mathbf{x})|^2\right] \geq \varepsilon$, then since $h, f$ have range $[0, R]$, we obtain that $\mathbb{E}_\mathcal{D}\left[R|h(\mathbf{x}) - f^*(\mathbf{x})|\right] \geq \mathbb{E}_\mathcal{D}\left[|h(\mathbf{x}) - f^*(\mathbf{x})|^2\right] \geq \varepsilon$ i.e., $\mathbb{E}_\mathcal{D}\left[|h(\mathbf{x}) - f^*(\mathbf{x})|\right] \geq \varepsilon/R$. Also, the optimal primary instance assignment for $\text{val}_1(\mathcal{I}, h)$ is also optimal for $\text{val}_2(\mathcal{I}, h)$ since the closest instance-prediction in a bag to the bag-label remains the same. Using this, along with the above analysis and substituting $\varepsilon/R$ for $\zeta$ along with the values of the other parameters we obtain the statement of Theorem 3.1.

## 4 HARDNESS OF LINEAR MIR

**Theorem 4.1.** *Let $\mathcal{F}$ be the class of all linear regressors over $\mathbb{R}^n$ for some $n \in \mathbb{Z}^+$. There is an absolute constant $C_2 > 0$ s.t. given an instance $\mathcal{I}$ of MIR whose bags are of size $\leq 2$ with bag-labels in $[-1, 1]$ such that there exists a $f^* \in \mathcal{F}$ such that $\text{val}_2(\mathcal{I}, f^*) = 0$, it is NP-hard to find $f \in \mathcal{F}$ s.t. $\text{val}_2(\mathcal{I}, f) \leq C_2 - \varepsilon$, for any constant $\varepsilon > 0$. In fact, one can take $C_2 = \frac{2}{100}\left(1 - \frac{1}{\sqrt{\pi}}\right)$. The result also holds when the bags are disjoint and therefore for injective MIR.*

The rest of this section describes the dictatorship test which is a key component of the proof, the rest of which is included in Appendix A. While the hardness reduction creates MIR instances with overlapping bags, in Appendix A.5 we show how to make the bags disjoint while retaining the hardness factor.

## 4.1 DICTATORSHIP TEST

For any positive integer $K$, let $\mathcal{J}_K$ an instance of MIR on 2-sized bags as follows. The underlying set of feature-vectors is $\{-1, 1\}^K$, we now define $\mathcal{J}_K$ as a distribution which samples a random 2-sized bag along with its label as follows:

1. Choose $\mathbf{x}^{(1)}$ uniformly at random from $\{-1, 1\}^K$ and define $\mathbf{x}^{(2)} = -\mathbf{x}^{(1)}$.
2. Sample $\sigma \leftarrow \{1, -1\}$ uniformly at random.
3. Output bag $B = \{\mathbf{x}^{(1)}, \mathbf{x}^{(2)}\}$ along with $\sigma$ as its label.

While we define $\mathcal{J}_K$ for convenience as a distribution over bags and their labels, the distribution is uniform over all possible $2^K$ sets $\{\mathbf{x}^{(1)}, \mathbf{x}^{(2)} = -\mathbf{x}^{(1)}\}$ and labels $\{-1, 1\}$ for each of them, in total we have $2^{K+1}$ bags. Note that in the above we treat a set of two feature-vectors with label $1$ and with label $-1$ as two distinct bags. We prove the following two properties of $\mathcal{J}_K$.

**Lemma 4.2** ((Completeness of $\mathcal{J}_K$)). *For any $i^* \in [K]$, the linear regressor $f^{(i)}(\mathbf{x}) = x_i$ admits a primary instance assignment $\Gamma$ such that its* $\mathrm{val}_2(\mathcal{J}_K, f^{(i)}, \Gamma) = 0$.

*Proof.* For any fixed $i \in [K]$, and any bag $B = \{\mathbf{x}^{(1)}, \mathbf{x}^{(2)}\}$ sampled by $D(\mathcal{J}_K)$, we have that $f^{(i)}(\mathbf{x}^{(1)}) = x_i^{(1)} = -x_i^{(2)} = -f^{(i)}(\mathbf{x}^{(2)}) \in \{-1, 1\}$. Thus, $\{f^{(i)}(\mathbf{x}^{(1)}), f^{(i)}(\mathbf{x}^{(2)})\} = \{-1, 1\}$ which are the two possible values of $\sigma$. Therefore, the choice of $\Gamma(B)$ to be $\mathbf{x}^a$ s.t. $f^{(i)}(\mathbf{x}^{(a)}) = \sigma$ yields that $\mathrm{val}_2(\mathcal{J}_K, f^{(i)}, \Gamma) = 0$. $\square$

### 4.1.1 Soundness of $\mathcal{J}_K$

Let $f(\mathbf{x}) = \langle \mathbf{c}, \mathbf{x} \rangle + c_0 = \sum_{i=1}^{K} c_i x_i + c_0$ denote a linear regressor over $\mathbb{R}^K$. We say that $f$ is $\tau$-regular if $\max_{i \in [K]} |c_i| \le \tau \|\mathbf{c}\|_2$. This subsection is devoted to proving the following soundness property of $\mathcal{J}_K$ for the above linear regressor.

**Lemma 4.3** ((Soundness of $\mathcal{J}_K$)). *There is an absolute constant $C_2 > 0$ such that for any small enough constant $\tau > 0$, if $f$ is $\tau$-regular then for any primary instance assignment $\Gamma$, $\mathrm{val}_2(\mathcal{J}_K, f, \Gamma) \ge C_2 - \tau$. In fact, one can take $C_2 = \frac{2}{100}\left(1 - \frac{1}{\sqrt{\pi}}\right)$.*

We assume the $\tau$-regularity condition of the above lemma and prove the lemma for the case $c_0 \le 0$, with the proof for $c_0 \ge 0$ being analogous.

Observe that the bag gets label either $1$ or $-1$ with equal probability. Therefore, it suffices to lower bound the probability that both $f(\mathbf{x}^{(1)})$ as well as $f(\mathbf{x}^{(2)})$ are far enough from $1$ (for $c_0 \ge 1$ the deviation from $-1$ is lower bounded), as the following lemma shows.

**Lemma 4.4.** *There are absolute constants $t_0 \in (0, 1/2], p_0 \in (0, 1]$ s.t.*

$$\Pr\left[\left|f(\mathbf{x}^{(1)}) - 1\right|, \left|f(\mathbf{x}^{(2)}) - 1\right| > t_0\right] \ge p_0 - 2.5\tau. \quad (9)$$

*In fact, the above is satisfied with $t_0 = 0.2$ and $p_0 = 1 - 1/\sqrt{\pi}$*

*Proof.* For convenience let $g(\mathbf{x}) := \langle \mathbf{c}, \mathbf{x} \rangle$ i.e., $f(\mathbf{x}) = g(\mathbf{x}) + c_0$. Consider a random bag $B$ sampled by $\mathcal{J}_K$. Define the random variable $\tilde{X} := g(\mathbf{x}^{(1)})$ and by construction we have that $g(\mathbf{x}^{(2)}) = -\tilde{X}$. Thus, we have

$$\min\left\{\left|f(\mathbf{x}^{(1)}) - 1\right|, \left|f(\mathbf{x}^{(2)}) - 1\right|\right\} > t_0$$
$$\Leftrightarrow \quad \min\{|X + c_0 - 1|, |-X + c_0 - 1|\} > t_0$$
$$\Leftrightarrow \quad \min\{|X - (1 - c_0)|, |-X - (1 - c_0)|\} > t_0$$
$$\Leftrightarrow \quad |X| \notin [\kappa - t_0, \kappa + t_0] \quad (10)$$

where $\kappa = 1 - c_0 \ge 1$ since $c_0 \le 0$. Now, $X = \sum_{i=1}^{K} c_i x_i$ where $x_i$ are i.i.d. uniform $\{-1, 1\}$ random variables, $i \in [K]$. Applying the Berry-Esseen theorem [Shevtsova, 2010] and using the regularity of $f$ we obtain that for any $t \in \mathbb{R}$

$$|\Pr[X > t] - \Pr[Z > t]| \le 0.57\tau \quad (11)$$

where $Z \sim N(0, \sigma^2)$ is a mean zero Gaussian with variance $\sigma^2 = \|\mathbf{c}\|_2^2$. Thus,

$$|\Pr[|X| \notin [\kappa - t_0, \kappa + t_0]]$$
$$- \Pr[|Z| \notin [\kappa - t_0, \kappa + t_0]]| \le 2.5\tau$$

To complete the proof, we need to lower bound the above probability term with $Z$. Since $Z$ is a mean-zero Gaussian and $\kappa - t_0 \ge \kappa - 1/2 > 0$, $\Pr[|Z| \notin [\kappa - t_0, \kappa + t_0]] = 1 - 2\Pr[Z \in [\kappa - t_0, \kappa + t_0]]$. Therefore, we need to upper bound $\Pr[Z \in [\kappa - t_0, \kappa + t_0]]$ by a constant strictly less than $1/2$. To do this we shall take $t_0 = 0.2 \le 0.2\kappa$ and consider two cases:

- $\sigma > 4\sqrt{2}\kappa/10$: In this case a straightforward integration over the segment $[\kappa - t_0, \kappa + t_0]$ of length $2t_0$ yields

$$\Pr[Z \in [\kappa - t_0, \kappa + t_0]]$$
$$\le \frac{2t_0}{\sigma\sqrt{2\pi}} \le \frac{2(0.2)(10)\kappa}{8\kappa\sqrt{\pi}} = \frac{1}{2\sqrt{\pi}}$$

- $\sigma \le 4\sqrt{2}\kappa/10$: In this case we have,

$$\Pr[Z \in [\kappa - t_0, \kappa + t_0]] \le \Pr\left[\frac{Z}{\sigma} \ge \frac{\kappa - t_0}{\sigma}\right]$$
$$\le \Pr\left[\frac{Z}{\sigma} \ge \sqrt{2}\right]$$

where we use the upper bound of $4\sqrt{2}\kappa/10$ on $\sigma$ and that $\kappa - t_0 \geq \kappa - 0.2\kappa = 0.8\kappa$. Now since $Z/\sigma \sim N(0,1)$, Prop 2.1.2 of [Vershynin, 2018] yields an upper bound for the RHS of above given by $\left(1/(\sqrt{2}\sqrt{2\pi})\right)\exp(-((\sqrt{2})^2/2) \leq 1/(2\sqrt{\pi})$.

Combining everything we complete the proof with $t_0 = 0.2$ and $p_0 = 1 - 1/\sqrt{\pi}$. $\qquad\square$

*Proof.* (of Lemma 4.3) Observe that a bag with the same two feature-vectors occurs with bag-label 1 as well as $-1$. Lemma 4.4 shows that the expected contribution to $\text{val}_2$ from bags with bag-label 1 is $(p_0 - 2.5\tau)t_0^2$. Since, bag-label 1 occurs half the time, we obtain the following lower bound on the $\text{val}_2(\mathcal{J}_K, f, \Gamma)$ for any $\Gamma$:

$$\frac{1}{2}\left(1 - \frac{1}{\sqrt{\pi}} - 2.5\tau\right)(0.2)^2 \geq \frac{2}{100}\left(1 - \frac{1}{\sqrt{\pi}}\right) - \tau \tag{12}$$

which completes the proof. $\qquad\square$

# 5 WEIGHTED ASSIGNMENT TRAINING

We describe our wtd-Assign model training method. Let $\mathcal{I}$ be an instance of injective MIR as defined in Sec. 2. Let $k_j$ ($j \in [m]$) be the size of the $j$th bag given by $B_j = \{\mathbf{x}_{ij} \mid i = 1, \ldots, k_j\}$, and $n = \sum_{j=1}^m k_j$ be the total elements with multiplicity of all the bags. Let $\mathcal{X}$ be the set of distinct feature-vectors in $\cup_{B \in \mathcal{B}}$. For each $\mathbf{x} \in \mathcal{X}$ let $J(\mathbf{x}) := \{(i,j) \mid \mathbf{x} = \mathbf{x}_{ij}\}$. Since each bag is a subset (i.e., with no multiplicities) each $J(\mathbf{x})$ has at most one tuple corresponding to any $j$.

**Predictor Model.** We train a real-valued model $M$ over the domain $\mathcal{X}$ i.e., $M : \mathcal{X} \to \mathbb{R}$.

**Trainable free variables.** We define $z_{ij} \in R$ to trainable variables for each $(i,j) \in \cup_{j=1}^m \{1, \ldots, k_j\} \times \{j\}$. Note that these are real-valued *free* variables which are not outputs from the predictor model $M$. Denote set of such variables as $Z$.

**Derived variables.** For each $z_{ij}$ there is a corresponding variable $u_{ij} := \text{Sigmoid}(z_{ij}) = 1/(1 + e^{-z_{ij}}) \in (0,1)$ denoting the the probability that $\mathbf{x}_{ij}$ is primary for bag $j$. Let the collection of all the $u$ variables be denoted by $U$.

**Loss Function.** Given the variables $U$, our first regularization loss term pushes each $u \in U$ to be either 0 or 1 using an entropic loss:

$$\mathcal{L}_{\text{SE}}(U) := \sum_{u \in U} (-u\log u - (1-u)\log(1-u)) \tag{13}$$

The second regularization loss term ensures that each bag has exactly one primary instance:

$$\mathcal{L}_{\text{prob}}(\mathcal{B}) := \sum_{j=1}^m \left| \sum_{i=1}^{k_j} u_{ij} - 1 \right| \tag{14}$$

The next one similarly makes sure that an instance is primary in at most one bag

$$\mathcal{L}_{\text{prim}}(\mathcal{X}) := \left| \max\left\{ \sum_{\mathbf{x} \in \mathcal{X}} \sum_{(i,j) \in J(\mathbf{x})} u_{ij}, 1 \right\} - 1 \right| \tag{15}$$

Lastly, we minimize the deviation of the bag-label prediction from the true bag-label using:

$$\mathcal{L}_{\text{bag}}(\mathcal{B}) := \sum_{j=1}^m L_{\text{bag}}\left( \sigma_j, \sum_{i=1}^{k_j} u_{ij} M(\mathbf{x}_{ij}) \right) \tag{16}$$

where $\mathcal{L}_{\text{bag}}$ is typically mase or the mean absolute error (mae). For convenience we will use $\mathcal{L}_{\text{SE}}(V)$ to denote the restriction of $\mathcal{L}_{\text{SE}}(U)$ to only those variables in $V \subseteq U$, and similarly for any $\mathcal{B}_0 \subseteq \mathcal{B}$, $\mathcal{L}_{\text{prob}}(\mathcal{B}_0)$ and $\mathcal{L}_{\text{bag}}(\mathcal{B}_0)$ are corresponding restrictions to the bags in $\mathcal{B}_0$ in which the summations in the RHS of (14) and (16) respectively are only over the bags in in $\mathcal{B}_0$. For convenience, $\mathcal{L}_{\text{prim}}(\mathcal{B}_0)$ is used to denote the restriction of (15) to only the bags $\mathcal{B}_0$ i.e., the summation is over only the instances $\mathbf{x}$ present in $\mathcal{B}_0$.

The combined wtd-Assign loss that we optimize is:

$$\mathcal{L}_{\text{WA}}(U, \mathcal{B}) = \lambda_1 \mathcal{L}_{\text{SE}}(U) + \lambda_2 \mathcal{L}_{\text{prob}}(\mathcal{B})$$
$$+ \lambda_3 \mathcal{L}_{\text{prim}}(\mathcal{B}) + \lambda_4 \mathcal{L}_{\text{bag}}(\mathcal{B}) \tag{17}$$

for some hyperparameters $\lambda_1, \lambda_2, \lambda_3, \lambda_4 \geq 0$.

**Minibatch based model training.** For a given set of hyperparameters $\{\lambda_t\}_{t=1}^4$, learning rate $\delta$, optimizer `optimizer`, and a minibatch size $q$ the method trains the predictor model $M$ along with the variables $Z$ as follows by doing the following for $N$ epochs and $K$ steps per epoch:

1. Sample a minibatch $S$ of $q$ bags $\mathcal{B}_S \subseteq \mathcal{B}$.
2. For each distinct $(i,j)$ s.t. $B_j \in \mathcal{B}_S$ and $i \in [k_j]$, use $u_{ij} := \text{Sigmoid}(z_{ij})$ along with the predictions $M$ of the model on the required subset of variables from $Z$ to compute $u_{ij}$, and let $U_S := \{u_{ij} \mid B_j \in \mathcal{B}_0, i \in [k_j]\} \subseteq U$.
3. Use the values in $U_S$ to compute $\mathcal{L}_{\text{SE}}(U_S)$, $\mathcal{L}_{\text{prob}}(\mathcal{B}_S)$ and $\mathcal{L}_{\text{bag}}(\mathcal{B}_S)$.
4. For each feature-vector $\mathbf{x}$ in the bags $\mathcal{B}_S$ compute $\{u_{ij} \mid (i,j) \in J(\mathbf{x})\}$ using $u_{ij} := \text{Sigmoid}(z_{ij})$ along with the predictions $M$ of the model on the required subset of variables from $Z$. Use these to compute $\mathcal{L}_{\text{prim}}(\mathcal{B}_S)$.
5. Using the required gradients of $\mathcal{L}_{\text{WA}}(U_S, \mathcal{B}_S)$ from (17) along `optimizer` and learning rate $\delta$, update the weights of the model $M$.

# 6 EXPERIMENTAL EVALUATION

We comparatively evaluate our wtd-Assign method on synthetic as well as real-world data.

**Baselines.** The following baselines are included as part of our experiments:

1. Instance-MIR (InsMIR [Ray and Craven, 2005]) in which all the feature-vectors in a bag are labeled with the bag-label and the model is trained on the resultant data. For overlapping bags, multiple copies of the same feature-vector with different labels are used.
2. Aggregation-MIR (AggMIR [Wang et al., 2008]) in which the feature-vectors in a bag are averaged into a single feature-vector which is assigned the bag label and the model is trained on this aggregated dataset.
3. Primary-MIR (PIR [Ray and Page, 2001]) which is an EM based method which iteratively selects and updates the best instance in a bag as primary and trains the model on the selected primary instances.
4. Balanced-Pruning MIR (BPMIR [Wang et al., 2008]) in which those instances in a bag are removed which are farthest from the median prediction over the non-pruned bags. This is a more sophisticated – as well as empirically better performing – of the pruning based methods (see [Wang et al., 2008]).

## 6.1 SYNTHETIC DATASET EXPERIMENTS

Our synthetic data is generated over $n = 32$ dimensional real-space, with $m = 10000$ bags of size $k = 2, 5$, and $10$ each using the following steps.

*Feature-vector generation:* $mk$ feature-vectors are initially sampled i.i.d. from $N(0, 1)^n$, and then partitioned into $m$ subsets of size $k$ each. For each of the 32 features and each of the $m$ subsets, a $k \times k$ Cholesky matrix is sampled and $k$-vector of feature values linearly transformed. Thus, within each subset, the values corresponding to each feature are made correlated. There is no correlation across features or across bags for the same feature.

*Bag generation:* We create overlapping bags by resampling them as follows. For each bag and each instance $\mathbf{x}$ in that bag we center a Gaussian with a temperature-tuned log-likelihood at $\mathbf{x}$ and sample an instance from $\mathcal{X}$ using the normalized weights assigned by the temperature-tuned Gaussian, and replace $\mathbf{x}$ with the sampled instance in that bag. The temperature parameter is useful in controlling the degree of overlap. We define the *overlap percentage* as the fraction of feature-vectors that are part of more than one bag.

*Label generation:* A quadratic regressor over $\mathbb{R}^n$ is constructed by sampling $n$ linear coefficients randomly from $[-1, 1]$ and the $n + \binom{n}{2}$ quadratic term coefficients randomly from $[-0.1, 0.1]$. The instance-labels are given by this regressor and for each bag a random instance is chosen as primary and the bag-label is equated to its label with additive i.i.d. $N(0, 1)$ noise. Note that once an instance is made primary for one bag it is removed from the primary instance

candidates for the subsequent bags, so that one instance is primary in at most one bag i.e., this is an injective MIR setting.

The train-dataset consists of 8000 bags and the validation and test sets each consist of 2000 primary instances and their labels.

**Model Training.** The model used for all baselines is an neural net with one hidden layer of size 1024 and relu activations. The output node is a linear sum. The Adam optimizer is used in all our experiments. The mini-batch size is a hyperparameter ranging from 100 to 1000 bags, which along with the learning rate and weight decay as well as the weights for various loss terms in wtd-Assign are tuned using a grid search.

**Results.** Table 1 shows the test mse scores of the various methods with bag sizes $k = 5, 10$ and different overlap percentages (refer to Appendix B for results with $k = 2$). We observe that wtd-Assign is the best performing across the different overlap percentages. For smaller overlap percentages, the performance of PIR and BPMIR are closer to wtd-Assign while they significantly worsen for larger overlaps. This is expected as wtd-Assign explicitly handles overlapping bags.

Our technique wtd-Assign as well as PIR and BPMIR implicitly track the primary instances in each bag. Using this, in Table 2 we also present the attribution accuracy of these methods on the training bags i.e., on what percentage of training bags is the predicted primary instance same as the true primary instance. We again see wtd-Assign performs the best with stable accuracy scores across the overlap percentages. PIR is clearly the second best while its performance decreases noticeably with increasing overlap.

| Overlap % → | 10 | 15 | 20 | 25 |
|---|---|---|---|---|
| | | $k = 5$ | | |
| InsMIR | 7.55 | 9.09 | 9.48 | 11.12 |
| AggMIR | 13.84 | 13.95 | 13.71 | 13.89 |
| PIR | 3.20 | 4.32 | 4.95 | 3.94 |
| BPMIR | 3.46 | 3.85 | 4.12 | 4.69 |
| wtd-Assign | **2.61** | **2.87** | **2.74** | **3.17** |
| | | $k = 10$ | | |
| InsMIR | 16.12 | 18.61 | 22.97 | 28.46 |
| AggMIR | 30.45 | 30.35 | 30.19 | 32.00 |
| PIR | 7.95 | 9.35 | 11.51 | 13.46 |
| BPMIR | 7.29 | 12.13 | 15.03 | 21.34 |
| wtd-Assign | **6.23** | **8.47** | **8.80** | **11.77** |

Table 1: Synthetic data ($k = 5, 10$): Test MSE

| Overlap % $\rightarrow$ | 10 | 15 | 20 | 25 |
|---|---|---|---|---|
| $k = 5$ | | | | |
| PIR | 43.21 | 43.46 | 38.95 | 40.02 |
| BPMIR | 19.36 | 20.71 | 20.00 | 20.00 |
| wtd-Assign | **52.60** | **47.48** | **54.76** | **49.75** |
| $k = 10$ | | | | |
| PIR | 24.00 | 23.60 | 21.90 | 20.60 |
| BPMIR | 12.60 | 13.70 | 12.00 | 11.70 |
| wtd-Assign | **24.51** | **24.20** | **25.30** | **24.10** |

Table 2: Synthetic data ($k = 5, 10$): Train Attribution Accuracy

## 6.2 REAL-WORLD DATASET EXPERIMENTS

We use the 1940 US Census Data [Steven Ruggles and Sobek, 2018][1] from which we use the following features:

- *Target* : WKSWORK1 - Number of weeks the person worked in the previous year
- *Numerical Features* : AGE - Age of the person
- *Categorical Features*: SEX - gender, MARST - marital status, CHBORN - number of children born to a woman in that year, SCHOOL - school attendance, EMPSTAT - employment status, OCC - primary occupation, IND - type of industry in which the person works
- *Aggregation Features*: STATEICP, COUNTYICP, CITY, CNTRY, REGION.

We use the aggregation features only to create $k$-sized bags with $k = 16$ and $k = 25$. The first step is to group-by the aggregation features to obtain groups of instances corresponding to each setting of those features. We sample $k$-sized bags independently from each such group. As overlaps are desired, we discard those groups with less than 50 instances. From any remaining group of size $s$ we randomly sample $\approx s/k$ bags randomly and we also include a fraction of the instances into the test and validation sets. For each training bag, its label is obtained from a randomly chosen primary instance, making sure by resampling that an instance is primary for at most one bag. In total, we obtain $\approx 78,000$ training bags and $\approx 26,000$ sized test and validation sets for $k = 16$ and $\approx 53,000$ training bags and $\approx 18,000$ sized test and validation sets for $k = 25$. The overlap percentage is around 40%. The categorical features are encoded as multi-hot in a 402 dimensional space so that the input dimension is 403.

The model architecture, optimizer and the training hyperparameters are same as in the synthetic data experiments (Sec. 6.1)

[1]https://usa.ipums.org/usa/ 1940CensusDASTestData.shtml

**Results.** The model trained on the fully supervised training data has a test mse of 178.0. On the other hand, Table 3 reports the corresponding scores for the different methods trained on bags. We observe that wtd-Assign performs the best, however PIR is only slightly worse while BPMIR also has comparable performance. On the other hand InsMIR is significantly worse while the loss on AggMIR make it unusable.

| Bag size $\rightarrow$ | 16 | 25 |
|---|---|---|
| InsMIR | 290.96 | 295.93 |
| AggMIR | 1028.14 | 1297.38 |
| PIR | 286.60 | 319.93 |
| BPMIR | 219.98 | 223.02 |
| wtd-Assign | **208.03** | **211.75** |

Table 3: US Census data: Test MSE

The experimental code is available at https://github.com/google-research/google-research/tree/master/mir_uai24.

## 7 CONCLUSION

Our work proves the first generalization error bounds for the multiple instance regression (MIR) problem in which the label of a bag is given by that of an (unknown) primary instance in the bag. Specifically, we show that optimizing the mse loss on i.i.d sampled bags yields a regressor which has low mse on the underlying instance distribution, with high probability over the sampled bags. We also prove the first inapproximability result for MIR: given an MIR instance with bounded labels which admits a linear regressor with primary instances which has zero mse bag-loss, it is NP-hard to find one which has bag-loss lower than some absolute constant. While our contributions develop a deeper theoretical understanding of the problem, from a practical standpoint we also propose a weighted assignment based model training method which naturally handles overlapping bags unlike previous works. Our experiments on synthetic and real-world datasets demonstrate the improvements provided by our method.

Future work can include generalization guarantees for non-iid MIR bags, as well as investigation from the computational learning perspective of non-linear regressors on MIR data.

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

# Generalization and Learnability in Multiple Instance Regression (Appendix)

**Kushal Chauhan**[1]    **Rishi Saket**[1]    **Lorne Applebaum**[2]    **Ashwinkumar Badanidiyuru**[2]    **Chandan Giri**[2]

**Aravindan Raghuveer**[1]

[1]Google Research, India , {kushalchauhan,rishisaket,araghuveer}@google.com
[2] Google, USA , {lapplebaum,ashwinkumarbv,chandangiri}@google.com

## A   HARDNESS REDUCTION FOR LINEAR MIR

### A.1   PRELIMINARIES

Our hardness result is via a reduction from the SMOOTH-LABEL-COVER problem defined below.

**Definition A.1.** An instance of SMOOTH-LABEL-COVER $\mathcal{L}(G(V, E), N, M, \{\pi^{e,v} \mid e \in E, v \in e\})$ consists of a regular connected (undirected) graph $G(V, E)$ with vertex set $V$ and edge set $E$. Every edge $e = (v_1, v_2)$ is associated with projection functions $\{\pi^{e,v_i}\}_{i=1}^{2}$ where $\pi^{e,v_i} : [M] \to [N]$. A vertex labeling is a mapping defined on $L : V \to [M]$. A labeling $L$ satisfies edge $e = (v_1, v_2)$ if $\pi^{e,v_1}(L(v_1)) = \pi^{e,v_2}(L(v_2))$. The goal is to find a labeling which satisfies the maximum number of edges.

The following theorem states the hardness of SMOOTH-LABEL-COVER and is proved in Appendix A of [Guruswami et al., 2016].

**Theorem A.2.** *There exists a constant $c_0 > 0$ such that for any constant integer parameters $Q, R \geq 1$, it is NP-hard to distinguish between the following two cases for a* Smooth Label Cover *instance $\mathcal{L}(G(V, E), N, M, \{\pi^{e,v} \mid e \in E, v \in e\})$ with $M = 7^{(Q+1)R}$ and $N = 2^R 7^{QR}$:*

- *(YES Case) There is a labeling that satisfies every edge.*
- *(NO Case) Every labeling satisfies less than a fraction $2^{-c_0 R}$ of the edges.*

*In addition, the instance $\mathcal{L}$ satisfies the following properties:*

- *(Smoothness) For any vertex $w \in V$, $\forall i, j \in [M]$, $i \neq j$, $\Pr_{e \sim w}[\pi^{e,w}(i) = \pi^{e,w}(j)] \leq 1/Q$, where the probability is over a randomly chosen edge incident on $w$.*
- *For any vertex $v$, edge $e$ incident on $v$, and any element $i \in [N]$, we have $|(\pi^{e,v})^{-1}(i)| \leq d := 4^R$; i.e., there are at most $d = 4^R$ elements in $[M]$ that are mapped to the same element in $[N]$.*
- *(Weak Expansion) For any $\delta > 0$, let $V' \subseteq V$ and $|V'| = \delta \cdot |V|$, then the number of edges among the vertices in $|V'|$ is at least $\delta^2 |E|$.*

Theorem 4.1 follows from the following hardness reduction and Theorem A.2.

**Theorem A.3.** *There exists a universal constant $C_2 \in (0, 1]$ s.t. for any $\varepsilon > 0$ there exists a polynomial time reduction from an SMOOTH-LABEL-COVER instance $\mathcal{L}$ with some parameters $Q$ and $R$ depending on $\varepsilon$ to an instance $\mathcal{I}$ of MIR with all bags of size $\leq 2$ and labels in $[-1, 1]$ s.t.*

- *(YES Case) If $\mathcal{L}$ is a YES instance then there exists a linear regressor $h^*$ and a primary instance assignment $\Gamma^*$ satisfying $\mathrm{val}_2(\mathcal{I}, h^*, \Gamma^*) = 0$.*
- *(NO Case) Id $\mathcal{L}$ is a NO instance then for all linear regressors $h$ and primary instance assignments $\Gamma$, $\mathrm{val}_2(\mathcal{I}, h, \Gamma) > C_2 - \varepsilon$.*

*The above holds with $C_2 = \frac{2}{100}\left(1 - \frac{1}{\sqrt{\pi}}\right)$.*

The rest of this section is devoted to proving Theorem A.3.

## A.2   HARDNESS REDUCTION FROM SMOOTH-LABEL-COVER

Let $Q, R$ be parameters of an SMOOTH-LABEL-COVER $\mathcal{L}$ from Theorem A.2, to be set later depending on $\varepsilon$ in Theorem A.3. We first create an intermediate instance $\tilde{\mathcal{I}}$ of MIR as follows. For each vertex in $V$ we have a block of $M$ coordinates i.e., $\boldsymbol{\mathcal{X}} = \mathbb{R}^{V \times [M]}$. For a vector $\mathbf{x} \in \boldsymbol{\mathcal{X}}$, let $\mathbf{x}_v \in \mathbb{R}^M$ denote its restriction of the $M$ coordinates corresponding to $v \in V$. The instance $\tilde{\mathcal{I}}$ is define by a distribution $D_{\tilde{\mathcal{I}}}$ which samples a random bag as follows:

1. Sample a vertex $v \in V$ uniformly at random.
2. Sample a bag-label pair $\left(\{\mathbf{x}^{(1)}, \mathbf{x}^{(2)}\}, \sigma\right)$ from $\mathcal{J}_M$ (see Sec. 4.1).
3. Define vectors $\tilde{\mathbf{x}}^{(1)}$ and $\tilde{\mathbf{x}}^{(2)}$ as follows:

$$\forall u \in V, \qquad \tilde{\mathbf{x}}_u^{(1)} = \begin{cases} \mathbf{x}^{(1)} & \text{if } u = v \\ \mathbf{0} & \text{otherwise.} \end{cases} \qquad \tilde{\mathbf{x}}_u^{(2)} = \begin{cases} \mathbf{x}^{(2)} & \text{if } u = v \\ \mathbf{0} & \text{otherwise.} \end{cases} \tag{18}$$

4. Output the bag $\left(\{\tilde{\mathbf{x}}^{(1)}, \tilde{\mathbf{x}}^{(2)}\}, \sigma\right)$

We now apply the folding transformation to obtain the final instance.

### A.2.1   Folding and Final Instance $\mathcal{I}$

For any edge $e = (u, v) \in E$ and element $j \in [N]$, define the vector $\mathbf{h}^{(e,j)} \in \mathbb{R}^{V \times [M]}$ as follows,

$$h_{w,i}^{(e,j)} = \begin{cases} 1 & \text{if } w = u \text{ and } i \in (\pi^{e,u})^{-1}(j) \\ -1 & \text{if } w = v \text{ and } i \in (\pi^{e,v})^{-1}(j) \\ 0 & \text{otherwise.} \end{cases}$$

Therefore, for any vector $\tilde{\mathbf{x}} \in \mathbb{R}^{V \times [M]}$,

$$\forall e = \{u, v\} \in E, \; j \in [N], \quad \tilde{\mathbf{x}} \perp \mathbf{h}^{(e,j)} \Leftrightarrow \sum_{i \in (\pi^{e,u})^{-1}(j)} \tilde{x}_{u,i} = \sum_{i' \in (\pi^{e,v})^{-1}(j)} \tilde{X}_{v,i'} \tag{19}$$

Define two subspaces $H$ and $F$ of $\mathbb{R}^{V \times [M]}$ as:

$$H := \text{span}(\mathbf{h}^{(e,j)} \mid e \in E, \; j \in [N]\}) \quad \text{and } F = H^{\perp} \tag{20}$$

i.e, $F$ is the orthogonal complement of $H$ in $\mathbb{R}^{V \times [M]}$

The final instance $\mathcal{I}$ is obtained by replacing each bag $\left(\{\tilde{\mathbf{x}}^{(1)}, \tilde{\mathbf{x}}^{(2)}\}, \sigma\right)$ with a bag $\left(\{\overline{\mathbf{x}}^{(1)}, \overline{\mathbf{x}}^{(2)}\}, \sigma\right)$, where $\overline{\mathbf{x}}^{(s)}$ is the projection of the vector $\tilde{\mathbf{x}}^{(s)}$ onto $F$ and represented using a orthonormal basis for $F$ ($s = 1, 2$). Thus, the entire instance $\mathcal{I}$ along with the expected linear regressor solutions reside in $F$.

## A.3   PROOF OF YES CASE

Suppose $\rho : V \to [M]$ is a labeling that satisfies all edges $E$ of $\mathcal{L}$. We shall first construct a solution for $\tilde{\mathcal{I}}$ with objective 0. Consider the vector $\tilde{\mathbf{c}} \in \mathbb{R}^{V \times [M]}$ where $\tilde{c}_{v,i} = 1$ if $i = \rho(v)$ and 0 otherwise, for all $v \in V$ and $i \in [M]$. Observe that for any $v \in V$: (i) $\tilde{\mathbf{c}}$ is an indicator vector in the $M$ coordinates corresponding to $v$, and (ii) the bags of $\tilde{I}$ after sampling $v$ are exactly those of $\mathcal{J}_M$ in those coordinates (with coordinates corresponding to $v' \neq v$ being set to zero). Thus, by the the completeness of $\mathcal{J}_M$ (Lemma 4.2) $f^*(\mathbf{x}) := \langle \tilde{\mathbf{c}}, \mathbf{x} \rangle$ has zero objective on the bags of $\tilde{\mathcal{I}}$.

Now, since $\rho$ is a satisfying assignment, $\tilde{\mathbf{c}}$ satisfies the condition on the LHS of (19) using which we obtain that $\tilde{\mathbf{c}} \perp H$. Therefore, for any $\tilde{\mathbf{x}} \in \mathbb{R}^{V \times [M]}$, $\langle \tilde{\mathbf{c}}, \tilde{\mathbf{x}} \rangle = \langle \overline{\mathbf{c}}, \overline{\mathbf{x}} \rangle$, where $\overline{\mathbf{c}}$ and $\overline{\mathbf{x}}$ are the projections of $\tilde{\mathbf{c}}$ and $\tilde{\mathbf{x}}$ onto $H^{\perp} = F$. Thus, the objective of $\overline{\mathbf{c}}$ on $\mathcal{I}$ is same as that of $\tilde{\mathbf{c}}$ on $\tilde{\mathcal{I}}$ which is 0.

## A.4 PROOF OF NO CASE

Suppose for a contradiction that there is a regressor $\overline{f}(\overline{\mathbf{x}}) = \langle \overline{\mathbf{c}}, \overline{\mathbf{x}} \rangle + c_0$ where $\overline{\mathbf{c}} \in F$. for which there is a primary instance assignment $\Gamma$ s.t. $\text{val}_2(\mathcal{I}, \overline{f}, \Gamma) < C_2 - \varepsilon$. Here we shall choose $C_2$ to be the constant from Lemma 4.3. Since it suffices to prove the soundness for small enough values of $\varepsilon$, we shall take $\varepsilon \le C_2/2$. For $v \in V$, let $\text{val}_2(\mathcal{I}, \overline{f}, \Gamma, v)$ be the objective restricted to only those bags corresponding obtained after sampling $v$ i.e., $D_{\tilde{\mathcal{I}}}$ conditioned on $v$. Therefore, $\text{val}_2(\mathcal{I}, \overline{f}, \Gamma) = \mathbb{E}_{v \in V}\left[\text{val}_2(\mathcal{I}, \overline{f}, \Gamma, v)\right]$. It is easy to see that there must be $(\varepsilon/(2C_2))$-fraction of the vertices $V' \subseteq V$ s.t. $\text{val}_2(\mathcal{I}, \overline{f}, \Gamma, v) \le C_2 - \varepsilon/2$ for each $v \in V'$, if not then by averaging $\text{val}_2(\mathcal{I}, \overline{f}, \Gamma) > (1 - \varepsilon/(2C_2))(C_2 - \varepsilon/2) > C_2 - \varepsilon$ which is a contradiction.

We now *unfold* $\overline{\mathbf{c}}$, rewriting it as $\mathbf{c} \in \mathbb{R}^{V \times [M]}$ which satisfies the folding constraints (19), and letting the corresponding regressor over $\mathbb{R}^{V \times [M]}$ be $\overline{f}(\mathbf{x}) = \langle \overline{\mathbf{c}}, \mathbf{x} \rangle + c_0$ for the intermediate instance $\tilde{\mathcal{I}}$. From the above we have that for each $v \in V'$, $\text{val}_2(\tilde{\mathcal{I}}, f, \Gamma, v) \le C_2 - \varepsilon/2$. Using our setting of $C_2$ as the constant from Lemma 4.3, we obtain that for each $v \in V'$, $\mathbf{c}_v \in R^M$ is *not* $(\varepsilon/2)$-regular where $\mathbf{c}_v$ is the restriction of $\mathbf{c}$ to only those coordinates corresponding $v$. Thus, the subsets $S_v := \{i \in [M] \mid |c_{v,i}| \ge (\varepsilon/2)\|\mathbf{c}_v\|_2\}$ and $R_v := \{i \in [M] \mid |c_{v,i}| \ge (\varepsilon/4)\|\mathbf{c}_v\|_2\}$ are non empty for each $v \in V'$. Furthermore by definition, $S_v \subseteq R_v$ and $|S_v| \le (4/\varepsilon^2)$ and $|R_v| \le 16/\varepsilon^2$ for each $v \in V'$.

Let us also define the subset $T_v := \{i \in [M] \mid |c_{v,i}| \ge (\varepsilon/(16d))\|\mathbf{c}_v\|_2\}$ where $d := 4^R$ is the parameter from Theorem A.2, so that $S_v \subseteq R_v \subseteq T_v$ and $|T_v| \le (16d/\varepsilon)^2$, for all $v \in V'$. Letting $E'$ be the edges of $\mathcal{L}$ induced by $V'$, we obtain from Theorem A.2 that $|E'| \ge (\varepsilon/(2C_2))^2|E|$. Call an edge $e = \{u, v\} \in E'$ *good* if $|\pi^{e,u}(T_u)| = |T_u|$ and $|\pi^{e,v}(T_v)| = |T_v|$. For any vertex $v \in V'$, the fraction of edges $e \in E$ incident on $v$ and violating $|\pi^{e,v}(T_v)| = |T_v|$ is at most $|T_v|^2/Q \le (16d/\varepsilon)^4/Q$ from Theorem A.2. We can count these for each of the vertices and remove them, thus, yielding the number of good edges to be at least $\Delta|E|$ where

$$\Delta \ge \left(\frac{\varepsilon}{2C_2}\right)^2 - \frac{2}{Q}\left(\frac{16d}{\varepsilon}\right)^4 \tag{21}$$

We now prove the following structural lemma for good edges.

**Lemma A.4.** *For any good edge $e = \{u, v\}$, $\pi^{e,u}(R_u) \cap \pi^{e,v}(R_v) \neq \emptyset$.*

*Proof.* Without loss of generality assume that $\|\mathbf{c}_v\|_2 \ge \|\mathbf{c}\|_2$. Since $S_v \neq \emptyset$, let $i_0 \in S_v$ and $j^0 = \pi^{e,u}(i_0)$. Furthermore, since $e$ is good we know that $(\pi^{e,v})^{-1}(j_0) \cap T_v = 1$. Thus,

$$\left|\sum_{i \in (\pi^{e,v})^{-1}(j_0)} c_{v,i}\right| \ge \left(\frac{\varepsilon}{2} - d\frac{\varepsilon}{16d}\right)\|\mathbf{c}_v\|_2 \ge \frac{7\varepsilon}{16}\|\mathbf{c}_v\|_2 \tag{22}$$

where $\frac{\varepsilon}{2}\|\mathbf{c}_v\|_2$ is the lower bound on $|c_{v,i_0}|$ since $i_0 \in S_v$, and $\frac{\varepsilon}{16d}\|\mathbf{c}_v\|_2$ is an upper bound on $|c_{v,i_0}|$ for $i \in (\pi^{e,v})^{-1}(j_0) \setminus \{i_0\}$ from $(\pi^{e,v})^{-1}(j_0) \cap T_v = 1$. Now, for a contradiction assume that $(\pi^{e,u})^{-1}(j_0) \cap R_u = \emptyset$. By the goodness of $e$ we already have $(\pi^{e,u})^{-1}(j_0) \cap T_u = 1$. Thus,

$$\left|\sum_{i \in (\pi^{e,u})^{-1}(j_0)} c_{u,i}\right| \le \left(\frac{\varepsilon}{4} + d\frac{\varepsilon}{16d}\right)\|\mathbf{c}_u\|_2 \le \frac{5\varepsilon}{16}\|\mathbf{c}_u\|_2 \tag{23}$$

However, (22) and (23) violate the folding constraint (19) for $\mathbf{c}$, thus completing the proof. $\square$

**Randomized Labeling.** Consider the following randomized labeling of $V'$: for each $v \in V'$ assign a label uniformly at random from $R_v$. From (21), Lemma A.4 and the upper bound of $16/\varepsilon^2$ for any $R_v$, $v \in V'$, we obtain that this randomized labeling satisfies in expectation at least $\Delta^* = (\varepsilon/4)^4\Delta|E|$ edges. We can choose the parameter $R$ in Theorem A.2 to be large enough and $Q \gg d$ to be large enough so that $\Delta^* > 2^{-c_0 R}$ which is a contradiction to the NO case of Theorem A.2. This completes the proof of the NO case.

## A.5 NON-OVERLAPPING BAGS

The bags in the instance $\mathcal{I}$ are overlapping, particularly since the dictatorship test $\mathcal{J}_M$ and therefore $\tilde{\mathcal{I}}$ creates multiple copies of the same bag with different bag label, and because the folding step may identify feature-vectors. To make the bags

of $\mathcal{I}$ disjoint, we do the following: independently for each bag (including copies) $B$ sample $\gamma \in (0, \varepsilon/2)$ u.a.r. and scale the bag-label as well as both the feature-vectors in that bag by $(1 - \gamma)$.

First, note that since the original bag-labels were $\{-1, 1\}$, and each feature-vector is primary for at least one bag in the YES case, none of the feature-vectors can be $\mathbf{0}$. This also holds in the NO case, otherwise one can easily distinguish the YES and NO cases, leading to P = NP. Thus, one may assume that none of the feature-vectors in $\mathcal{I}$ are $\mathbf{0}$. Now, observe that since the scaling factor is independently sampled for each bag from a continuous range and the number of bags are finite, with probability $1$ over the choice of the scaling factors any two feature-vectors from two different bags will have different lengths, and therefore the bags are pairwise disjoint. Clearly, a perfect linear regressor (i.e., with zero loss) remains one since the the bag-label is scaled with the same factor as the bag feature-vectors, so the YES case is preserved. For the NO case, observe that this can reduce the loss by a factor of at most $(1 - \varepsilon/2)^2$ therefore the lower bound on the loss remains $C_2 - O(\varepsilon)$.

# B   SYNTHETIC DATA EXPERIMENTS WITH $k = 2$

We observe that for tiny bag sizes ($k = 2$), PIR is able to better identify prime instances correctly, resulting in better performance than wtd-Assign. For larger bag sizes though, our wtd-Assign method performs the best (Table 1).

| Overlap % → | 10 | 15 | 20 | 25 |
|---|---|---|---|---|
| | $k = 2$ | | | |
| InsMIR | 2.90 | 3.12 | 3.43 | 3.76 |
| AggMIR | 4.49 | 4.28 | 4.15 | 4.22 |
| PIR | **1.21** | **1.21** | **1.21** | **1.21** |
| BPMIR | 1.95 | 1.91 | 2.13 | 2.02 |
| wtd-Assign | 1.38 | 1.33 | 1.34 | 1.36 |

Table 4: Synthetic data ($k = 2$): Test MSE

| Overlap % → | 10 | 15 | 20 | 25 |
|---|---|---|---|---|
| | $k = 2$ | | | |
| PIR | **84.32** | **84.46** | **84.48** | **84.60** |
| BPMIR | 50.30 | 51.08 | 50.18 | 50.20 |
| wtd-Assign | 76.51 | 76.39 | 79.73 | 77.55 |

Table 5: Synthetic data ($k = 2$): Train Attribution Accuracy