# OpenReview forum: "Generalization and Learnability in Multiple Instance Regression"
_auai.org/UAI/2024/Conference — UAI 2024 poster_

### Official Review · Reviewer_zgTm · 2024-03-04

**Q2-1 Originality-Novelty:** 3
**Q2-2 Correctness-Technical Quality:** 3
**Q2-5 Clarity Of Writing:** 2

**Q1 Summary And Contributions:**

The authors analyze multiple instance regression (MIR). They first provide a generalization property, showing that good MIR performance leads to good individual instance regression performance. On the negative side, the authors furthermore show that finding optimal linear regressors in the MIR setting is NP-hard, even when allows for an approximation error. Finally the authors propose a novel model training method for MIR and compare their proposal against other methods on synthetic and real world data, showing an overall good performance.

**Q2-3 Extent To Which Claims Are Supported By Evidence:**

3: Good: the main claims are supported by convincing evidence (in the form of adequate experimental evaluation, proofs, (pseudo-)code, references, assumptions).

**Q2-4 Reproducibility:**

3: Good: key resources (e.g. proofs, code, data) are available and key details (e.g. proofs, experimental setup) are sufficiently well-described for competent researchers to confidently reproduce the main results.

**Q3 Main Strengths:**

The main results are non-trivial extensions of the previous literature.
 The first main result is a generalization property of a good MIR predictor: any MIR predictor with good performance on the MIR task, also has a good control on the individual instance task. The paper does a decent job in navigating the reader through the individual steps of proving this result.
The second main result is an extension on a hardness result of MIR: the authors show that even finding a good approximation of a linear regressor is NP-hard.

**Q4 Main Weakness:**

I consider the main weaknesses of the paper to be the language, part of the presentation, and choice of content. Until Section 4 my only main complaint is that the text contains many language, grammar and typesetting problems. Given the extensive explanation of the generalization result, the section on the hardness result is surprisingly short. While some points are covered in overview, the section is too brief when compared with the previous explanations. But even more surprising to me is the inclusion of the novel method to tackle the MIR setting. The method seems unrelated to the theoretical findings and is with that very detached from the rest.

**Q5 Detailed Comments To The Authors:**

- Theorem 1.1 (informal): not easy to follow as the notation is not introduced yet. Also, what is the O(1) notation?

- The same for Section 1.2: it would be better to first explain the notation in detail.

- Beginning of 3.2: I don't see how the statement follows from 17.1 of Anthony and Bartlett. The version I could find contains a statement that bounds the generalization gap (bounds the probability that the true and empirical error of any hypothesis exceeds a threshold). Furthermore the statement is for mappings into the interval [0,1]

**Q9 Complying With Reviewing Instructions:**

Yes

---

> ### Author Rebuttal · Authors · 2024-04-05
>
> We thank the Reviewer for their appreciation of the paper's contributions as well as their helpful feedback and comments. We address their concerns point-wise below.
>
> *language, grammar and typesetting problems.*
> Authors: We sincerely regret these editorial issues that have crept into the submission. We assure the Reviewer that the submission will be proof read thoroughly and all these issues along with any typos, will be corrected in the final version.
>
> *Theorem 1.1 (informal): not easy to follow as the notation is not introduced yet. Also, what is the O(1) notation? The same for Section 1.2: it would be better to first explain the notation in detail.*
> Authors: We will explain and simplify the notation for ease of understanding of the informal theorem statements (Theorems 1.1 and 1.2), and also for the overview in Section 1.2. The $O(.)$ is the order notation and $O(1)$ is used to denote some constant without explicitly defining it, for ease of notation.
>
> *Beginning of 3.2: I don't see how the statement follows from 17.1 of Anthony and Bartlett ... Furthermore the statement is for mappings into the interval [0,1]*
> Authors: The labeling is given by $f^*$ and therefore the true error of $h$ is $\mathbb{E}\_{\mathcal{D}}[|h(\mathbf{x}) - f^*(\mathbf{x})|]$. The empirical error is $\mathbb{E}\_{\mathbf{x} \in  \mathcal{Z}}[|h(\mathbf{x}) - f^*(\mathbf{x})|]$ where $\mathcal{Z}$ is the $mk$ sized sample, and the expectation is over $\mathbf{x}$ sampled uniformly at random from $\mathcal{Z}$. Therefore, given that $\mathbb{E}\_{\mathcal{D}}[|h(\mathbf{x}) - f^*(\mathbf{x})|]  \geq \hat{\zeta}$ for all $h$ in $\mathcal{F}\_{\textnormal{err}}$, the condition of equation (5) follows by an upper bound of $\hat{\zeta}/2$ on the difference between true and empirical errors for the class $\mathcal{F}\_{\textnormal{err}}$ given by Theorem 17.1 of Anthony and Bartlett. Since the mappings in the latter are to $[0,1]$ instead of [0, R] in our case, we apply Theorem 17.1 of Anthony and Bartlett with $f^*/R$ as the labeling and $\overline{\mathcal{F}}\_{\textnormal{err}}$ := {$h/R : h \in \mathcal{F}\_{\textnormal{err}}$} as the function class. This amounts to taking $\hat\zeta/(2R)$ as the error $\epsilon$ in Theorem 17.1 of Anthony and Bartlett. Observing that $N\_1(\hat\zeta/(32R), \overline{\mathcal{F}}\_{\textnormal{err}}, 2mk) = N\_1(\hat\zeta/(32), \mathcal{F}\_{\textnormal{err}}, 2mk)$ completes the argument.
> For more clarity however, we shall add the above explanation.
>
> $ $
>
> *the section on the hardness result is surprisingly short.*
> Authors: Due lack of space, we deferred the proof of the hardness result to Appendix A. In the final version of the paper, we shall utilize the additional two pages to move the dictatorship test and its analysis, which are key parts of the proof (Appendices A.2 and A.3 currently) into the main paper.
>
> *the novel method to tackle the MIR setting .. seems unrelated to the theoretical findings*
> Authors: We believe our novel algorithm for training models for MIR builds upon our theoretical results. Our generalization error bound provides evidence that minimizing the bag-level MSE loss is a good way of learning the instance-labeling function. This justifies the optimization approach of our Weighted Assignment method which explicitly minimizes the bag-level loss by optimizing over the regressor and the prime instance assignment, so that the optimal regressor generalizes well on the instance-level data.
> Our hardness result also treats the MIR problem as an optimization one and proves intractability results for it.
> The effectiveness of our algorithm - inspired by the generalization bound - is demonstrated by the experiments. This also shows the practical implications of the theoretical results, which we believe is a useful contribution to this field of research.
>
> $ $
>
> We commit to open-sourcing our experimental code along with the final version of the paper.

---

### Official Review · Reviewer_A8Uq · 2024-03-07

**Q2-1 Originality-Novelty:** 3
**Q2-2 Correctness-Technical Quality:** 3
**Q2-5 Clarity Of Writing:** 3

**Q1 Summary And Contributions:**

This paper studies the theoretical aspects of multi-instance regression. It provides generalization bounds when bags are i.i.d. using pseudo-dimensions as tool. An interesting finding in this result is that the distribution of instances do not need to be specified. It also proves hardness results on multi-instance regression under the commonly assumed one instance responsible for the bag target setting. A practical algorithm is then proposed for the setting of overlapping bags, e.g., some instances exist across multiple bags, which achieve state-of-the-art when compared to existing multi-instance regression techniques under the overlapping bags setting.

**Q2-3 Extent To Which Claims Are Supported By Evidence:**

3: Good: the main claims are supported by convincing evidence (in the form of adequate experimental evaluation, proofs, (pseudo-)code, references, assumptions).

**Q2-4 Reproducibility:**

2: Fair: key resources (e.g. proofs, code, data) are unavailable but key details (e.g. proof sketches, experimental setup) are sufficiently well-described for an expert to confidently reproduce the main results.

**Q3 Main Strengths:**

1. This work obtained new theoretical results for the challenging problems of multiple instance regression.
2. The theoretical results are supported with experiments.

**Q4 Main Weakness:**

1. Impact of the new theoretical results have a limited impact on practical algorithms as multi-instance regression receives much less attention than multi-instance classification in practice.
2. The presentation of this work could use some improvement. Although the logic is well-flowed, the writing could be difficult to understand at times.

**Q5 Detailed Comments To The Authors:**

In the introduction, it is not immediately clear why this work focuses on overlapping bags and the concept of overlapping bags is not clearly discussed. There could be better discussions between the informal theorem 1.2 and the practical standpoint.

The authors state that existing work requires bags are pairwise disjoint, i.e., one instance can only exist in one bag. However, this seems not to be the case at least in many practical MIL algorithms.

Some grammatical errors/typos exist which hinder readability. For example:
(1) "the optimum is strictly less that some absolute constant" should be "less than"?
(2) "iid" may be better to be spelled as "i.i.d." to indicate its abbreviation
(3) "w.h.p" could be spelled out for at least once to with high probability
(4) "shown to be NP-hard in general by Ray and Page [2001] who however left..." could use a comma before who
and many more (I have too many papers to review from UAI and ICML and this proofreading is really the authors' responsibility).

**Q9 Complying With Reviewing Instructions:**

Yes

---

> ### Author Rebuttal · Authors · 2024-04-05
>
> We are grateful to the Reviewer for their encouraging evaluation of the paper, and for their constructive feedback and comments. We address their concerns below.
>
> *Impact of the new theoretical results .. multi-instance regression receives much less attention.*
> Authors: We believe that MIR is of current practical relevance and any theoretical insights can impact the design and analysis of new techniques for MIR and are therefore interesting. In particular, the generalization error bound is the first such for the MIR problem - it shows that optimizing the bag-level MSE loss provably (in the case of random bags) learns the instance-wise labeling. This justifies the algorithmic approach of finding the prime instance assignment and the regressor to optimize the bag-level loss.
> Note that there are efficient algorithms to find the optimum linear regressor in the usual instance-wise (i.e., unit sized bags) case. In particular, given a set of point-label pairs which admit a perfect linear regressor one can effciently find one by solving a system of equations. Our hardness result shows that this problem becomes hard to even approximately solve when we have bags of size 2, and rules out a straightforward application of the simple techniques of the instance-wise case.
>
> To demonstrate relevance of MIR, these are some recent applications of MIR across multiple areas: in a novel deployment of MIR, the work of (Serafini et al. 2022) used it to model electrical load disaggregation. In the biological domain, the work of (Park et al. 2020) uses MIR to model the continuous response of bags of neoantigens. For image quality assessment where each image patch has a probability of being prime, the work of (Liang at al. 2021) applied an MIR approach to train a CNN. A different image analysis task -  estimating facial age from images - has also been tackled using MIR techniques (Liu et al. 2019).
>
> We will add the above use-cases of MIR as well as the references below:
>
> (Serafini et al. 2022) L. Serafini, G. Tanoni, E. Principi, S. Spinsante and S. Squartini, "A Multiple Instance Regression Approach to Electrical Load Disaggregation," 2022 30th European Signal Processing Conference (EUSIPCO), 2022, pp. 1666-1670.
>
> (Park et al. 2020) Park S, Wang X, Lim J, Xiao G, Lu T, Wang T. “Bayesian multiple instance regression for modeling immunogenic neoantigens”. Statistical Methods in Medical Research. 2020;29(10):3032-3047.
>
> (Liang at al. 2021) Dong Liang, Xinbo Gao, Wen Lu, Jie Li, “Deep blind image quality assessment based on multiple instance regression”, Neurocomputing, Volume 431,2021,Pages 78-89.
>
> (Liu et al. 2019) Liu, J., Qiao, R., Li, Y., Li, S., “Witness detection in multi-instance regression and its application for age estimation”. Multimed Tools Appl 78, 33703–33722 (2019).
>
> $ $
>
> *Condition of pairwise disjoint bags in previous work*
> Authors: We state in Section 1.1 (Previous Related Work) that much of the previous work requires that the bags are pairwise disjoint. We do mention however that there are some previous techniques that are applicable for overlapping bags, for e.g. aggregated-MIR and pruning-MIR. Other MIR techniques, such as instance-MIR - where each instance in a bag is assigned the bag label - are not well defined for the overlapping bags case.
>
> *it is not immediately clear why this work focuses on overlapping bags*
> Authors: We have mentioned, in the second to last paragraph before Section 1.1, our motivation for studying overlapping bags along with previously studied scenarios in which overlaps between bags arise. The above cited recent work (Serafini et al. 2022) also considers a novel scenario for MIR in which overlapping bags arise. We do however propose to move this part to the beginning of the introduction so that our motivation to study overlapping bags appears earlier in the paper.
>
> *Writing/Grammatical errors, the writing could be difficult to understand at times.*
> Authors: We are grateful to the Reviewer for pointing out the typos, which we will correct along with any others, as well as spell out the abbreviations for the first time they are used. We sincerely regret these editorial issues that have crept into the submission. We assure the Reviewer that the submission will be proof read thoroughly and the writing improved in the final version.
>
> $ $
>
> We commit to open-sourcing our experimental code along with the final version of the paper.

---

### Official Review · Reviewer_Npgv · 2024-03-08

**Q2-1 Originality-Novelty:** 3
**Q2-2 Correctness-Technical Quality:** 3
**Q2-5 Clarity Of Writing:** 4

**Q1 Summary And Contributions:**

The article considers the problem of multiple instance regression (MIR) in which the observations are grouped, only one output value per group is known, and this output value matches the output value of one observation (so called primary instance) in a group. The article introduces a new training method that for MIR that handles overlapping bags. According to the experiments, the method has lower test MSE than the earlier MIR methods. The article proves a generalization bound for MIR and establishes a strong inapproximability bound for linear regression in the MIR setting.

**Q2-3 Extent To Which Claims Are Supported By Evidence:**

3: Good: the main claims are supported by convincing evidence (in the form of adequate experimental evaluation, proofs, (pseudo-)code, references, assumptions).

**Q2-4 Reproducibility:**

3: Good: key resources (e.g. proofs, code, data) are available and key details (e.g. proofs, experimental setup) are sufficiently well-described for competent researchers to confidently reproduce the main results.

**Q3 Main Strengths:**

The article contains strong theoretical results. The experimental results demonstrate that the proposed method yields better results than the earlier approaches.

**Q4 Main Weakness:**

The solved problem (MIR) seems pretty niche, and I am not sure about its timeliness or the relevance to the field.

**Q5 Detailed Comments To The Authors:**

This is a minor detail, but when the names of the authors are used as a part of the sentence they should placed outside of the brackets, e.g., "Our focus is multiple instance regression (MIR) introduced by Ray and Page [2001]" instead of "Our focus is multiple instance regression (MIR) introduced by [Ray and Page 2001]".

**Q9 Complying With Reviewing Instructions:**

Yes

---

> ### Author Rebuttal · Authors · 2024-04-05
>
> We thank the Reviewer for their positive evaluation of the paper's contributions along with their helpful comments and feedback. In the following we address their concerns.
>
> *timeliness or the relevance of MIR*
> Authors: We have included in the Introduction (Sec. 1) references to previous papers which have used MIR for applications such as remote sensing and crop yield prediction.
> In addition, here we list out some recent applications of MIR across multiple areas. In a novel deployment of MIR, the work of (Serafini et al. 2022) used it to model electrical load disaggregation. In the biological domain, the work of (Park et al. 2020) uses MIR to model the continuous response of bags of neoantigens. For image quality assessment where each image patch has a probability of being prime, the work of (Liang at al. 2021) applied an MIR approach to train a CNN. A different image analysis task -  estimating facial age from images - has also been tackled using MIR techniques (Liu et al. 2019).
>
> We will add the above use-cases of MIR as well as the references below:
>
> (Serafini et al. 2022) L. Serafini, G. Tanoni, E. Principi, S. Spinsante and S. Squartini, "A Multiple Instance Regression Approach to Electrical Load Disaggregation," 2022 30th European Signal Processing Conference (EUSIPCO), 2022, pp. 1666-1670.
>
> (Park et al. 2020) Park S, Wang X, Lim J, Xiao G, Lu T, Wang T. “Bayesian multiple instance regression for modeling immunogenic neoantigens”. Statistical Methods in Medical Research. 2020;29(10):3032-3047.
>
> (Liang at al. 2021) Dong Liang, Xinbo Gao, Wen Lu, Jie Li, “Deep blind image quality assessment based on multiple instance regression”, Neurocomputing, Volume 431,2021,Pages 78-89.
>
> (Liu et al. 2019) Liu, J., Qiao, R., Li, Y., Li, S., “Witness detection in multi-instance regression and its application for age estimation”. Multimed Tools Appl 78, 33703–33722 (2019).
>
>
> $ $
>
> *Citation format: when the names of the authors are used as a part of the sentence they should placed outside of the brackets*
> Authors: We accept this suggestion and will correct citation grammar/format  accordingly.
>
> $ $
>
> We also commit to open-sourcing our experimental code along with the final version of the paper.

---

### Official Review · Reviewer_29XE · 2024-03-21

**Q2-1 Originality-Novelty:** 3
**Q2-2 Correctness-Technical Quality:** 3
**Q2-5 Clarity Of Writing:** 3

**Q1 Summary And Contributions:**

The authors mainly prove two new theoretical results for the Multiple Instance Regression (MIR) problem, with a practical weighted assignment training model in the case of overlapping bags. The first Theorem states a generalization error bound when bags are i.i.d. drawn, involving both the MSE loss on the sampled bags and the MSE instance-wise loss. The second Theorem states the impossibility (NP-hard) to achieve minimal bag-wise error if all bags are of size at most two. The authors propose a scaling perturbation trick to obtain pairwise disjoint from overlapping bags.
The proposed methodology is empirically evaluated with numerical experiments using both synthetic and real datasets.

**Q2-3 Extent To Which Claims Are Supported By Evidence:**

4: Excellent: all claims are supported by very convincing evidence (in the form of comprehensive experimental evaluation, rigorous mathematical proofs, detailed (pseudo-)code, precise references, well-motivated and realistic assumptions) and the authors deliver what they promise.

**Q2-4 Reproducibility:**

3: Good: key resources (e.g. proofs, code, data) are available and key details (e.g. proofs, experimental setup) are sufficiently well-described for competent researchers to confidently reproduce the main results.

**Q3 Main Strengths:**

The contributions are clearly exposed and compared to the literature, and essentially to the work of Ray and Page (2001). The effort to disentangle the technical proofs is very valuable (informal theorems, discussions and dedicated paragraphs on the required proof techniques).
The motivation and applicability of such models in real-world contexts are both convincing. The experimental section is, in that perspective, interesting and well organized.

**Q4 Main Weakness:**

The following points are subject to the possibility of having missed or misunderstood some information.

* Clarify section 2: the introduction of the mathematical objects lack rigor: assumptions on $\mathcal{X}$, the probability space, the involved random variables (eg regularity), definition of the covering number (it depends on the probability function of the random sample of observations, thus is random too), $N_{\infty}$ is not defined, the last sentence of section 2.1 should be more carefully stated.
* For section 2.2, the authors should also refer to the seminal book [Weak Convergence and Empirical Processes, Aad W. Vaart, Jon A. Wellner, Springer Series in Statistics, 1996).

* On the minibatch based training: how to choose the size $q$ wrt the total sample size or is it considered fixed in practice? Is there a stopping rule after step 5?

* Experiments: Authors should provide an online repository for their related code to promote replicable results. The numerical results should be presented with at least, their related standard deviation.

**Q5 Detailed Comments To The Authors:**

* It would be interesting to empirical analyze the impact on the algorithmic performance when making the size of the bag vary ($k$), and especially when it tends to $2$.
* A more detailed presentation of the experiments and possibly adding results on multiple real datasets could improve the statements of the paper.

Some additional comments:
* Missing definitions: assumptions on $\mathcal{X}$ (introduction), $c_i$ (page 3),  $\varepsilon$  (Th. 1.1), $E_{x\in \mathcal{Z}}$ (section 3)
* Section 3.1.2 seems incomplete and the statement is too loosely formulated.
* The series of Lemmas in section 3 should be stated with the related assumptions and definition of the parameters involved.
* The notation chosen to define the optimized $\mathcal{O}$ can be confused with convergence-related classical notation.

As a last note, I would have appreciated if the level of details of the proofs for each Theorem was balanced in the main corpus of the article. In the present exposition: Th. 1.1 is proved (with some loose statements) in the main corpus, while Th. 1.2 is only stated in the main corpus and proved in the appendix.

**Q9 Complying With Reviewing Instructions:**

Yes

---

> ### Author Rebuttal · Authors · 2024-04-05
>
> We are grateful to the Reviewer for their encouraging evaluation of the paper, as well as their constructive feedback and suggestions. We address their comments below.
>
> *Clarify section 2*
> Authors: The only assumption we have on $\mathcal{X}$ is that it is a subset of $\mathbb{R}^d$  for some positive integer $d$, there is some probability distribution $\mathcal{D}$ over it, and $\mathcal{F}$ is a class of real-valued functions (regressors) over $\mathcal{X}$. For any $\xi > 0$ and $N \in \mathbb{Z}^+$, the covering number $N_p(\xi, \mathcal{F}, N)$ is a fixed non-random quantity (Sec. 10.4 of [Anthony and Bartlett, 2009]).
> In the first paragraph of Sec. 2.2, $N\_p(\xi, \mathcal{F}, N)$ is defined corresponding to the $\ell_p$ metric for any $p$, and in particular it holds for $p = \infty$. The last line of Sec. 2.1 will be rephrased to be more formal.
> We will add these clarifications Section 2, add relevant references from  [Vaart and Wellner 1996] (thank you for pointing out) and make the mathematical preliminaries more formally defined.
>
> *minibatch based training*
> Authors: The minibatch size is treated as a hyperparameter along with the learning rate, and optimized using a grid search.  We don’t early stop our models. We use the model checkpoint with best validation performance, and report its MSE on the test set for our comparisons.
>
> *Experiments*
> Authors: We will open-source our code along with the final version of the paper, and add the standard deviations of the experimental results. On varying bag size $k$, especially when $k\rightarrow 2$, we will add the following synthetic data experiments, reporting the test MSE-error.
>
> $ $
>
> $k=2$
> | Overlap % -> | 5    | 10   | 15   | 20   | 25   |
> |--------------|------|------|------|------|------|
> | InsMIR       | 2.60 | 2.90 | 3.12 | 3.43 | 3.76 |
> | AggMIR       | 4.25 | 4.49 | 4.28 | 4.15 | 4.22 |
> | PIR          | 1.22 | 1.21 | 1.21 | 1.21 | 1.21 |
> | BPMIR        | 1.79 | 1.95 | 1.91 | 2.13 | 2.02 |
> | wtd-Assign   | 1.34 | 1.38 | 1.33 | 1.34 | 1.36 |
>
> $ $
>
> $k=5$
> | Overlap % -> | 5     | 10    | 15    | 20    | 25    |
> |--------------|-------|-------|-------|-------|-------|
> | InsMIR       | 6.63  | 7.55  | 9.09  | 9.48  | 11.12 |
> | AggMIR       | 13.77 | 13.84 | 13.95 | 13.71 | 13.89 |
> | PIR          | 2.81  | 3.20  | 4.32  | 4.95  | 3.94  |
> | BPMIR        | 3.38  | 3.46  | 3.85  | 4.12  | 4.69  |
> | wtd-Assign   | 2.17  | 2.61  | 2.87  | 2.74  | 3.17  |
>
> We see than on $k=2$ PIR performs better (as there is less uncertainty in the prime instances), while on $k=5$ our wtd-Assign method performs best.
> We will also add experiments using a larger sample of the US Census data on which we observe our method performing the best.
> Bag size = 16: 78k train bags, 26k val and test instances
> Bag size = 25: 53k train bags, 18k val and test instances
> | Bag size -> | 16      | 25      |
> |-------------|---------|---------|
> | InsMIR      | 290.96  | 295.93  |
> | AggMIR      | 1028.14 | 1297.38 |
> | PIR         | 286.60  | 319.93  |
> | BPMIR       | 219.98  | 223.02  |
> | wtd-Assign  | 208.03  | 211.75  |
>
> *Additional comments*
> Authors: The  notation $\mathbb{E}\_{\mathbf{x} \in \mathcal{Z}}$ in Sec. 3 denotes the expectation over uniformly sampled point $\mathbf{x}$ from $\mathcal{Z}$, and we will state this formally. The $c_i$ and $\mathbf{c}$ on page 3 should be $r_i$ and $\mathbf{r}$, and $\varepsilon$ in Theorem 1.1. should be $\varepsilon\_{\textnormal{MIR}}$ - we will correct these typos along with any others. We will make the proofs in Section 3 more formal and explicitly define the parameters used, and change $\mathcal{O}$ to $\texttt{optimizer}$ to avoid confusion.
>
> *details of the proofs for each Theorem to be balanced*
> Authors: Due lack of space, we deferred the proof of the hardness result (Theorem 1.2) to Appendix A. In the final version of the paper, we shall utilize the additional two pages to move the dictatorship test and its analysis, which are key parts of the proof (Appendices A.2 and A.3 currently) into the main paper.

---

### Official Review · Reviewer_zA88 · 2024-03-23

**Q2-1 Originality-Novelty:** 2
**Q2-2 Correctness-Technical Quality:** 3
**Q2-5 Clarity Of Writing:** 2

**Q1 Summary And Contributions:**

The submitted draft introduces novel findings in the field of Multiple Instance Regression (MIR) by addressing generalization and learnability aspects. It presents theoretical proofs, including the hardness of linear MIR and a weighted assignment model training method to handle overlapping instances in bags. The paper offers insights into the complexities of MIR, theoretical rigor, and practical implications for real-world applications.

**Q2-3 Extent To Which Claims Are Supported By Evidence:**

3: Good: the main claims are supported by convincing evidence (in the form of adequate experimental evaluation, proofs, (pseudo-)code, references, assumptions).

**Q2-4 Reproducibility:**

3: Good: key resources (e.g. proofs, code, data) are available and key details (e.g. proofs, experimental setup) are sufficiently well-described for competent researchers to confidently reproduce the main results.

**Q3 Main Strengths:**

Strenghts:

Novelty: The paper presents novel findings on the generalization and learnability aspects of Multiple Instance Regression (MIR), a less explored area compared to Multiple Instance Learning (MIL) .

Comprehenseive Theoretical analysis: The paper provides theoretical clear proofs and theorems, such as the hardness of linear MIR and weighted assignment model training, demonstrating a strong theoretical foundation.

Algorithmic contribution: The proposed weighted assignment model training method for MIR addresses the challenge of handling overlapping instances in bags, showcasing innovative approaches to tackle real-world problems .

Reasonable Empirical Analysis.

**Q4 Main Weakness:**

Relevance and Practical Implications: the paper and the results are poorly motivated. I am finding it hard to appreciate the importance of this problem. There are almost no recent cited works (only 1 paper from last 3 years from the citations), which bothers me about the importance of the problem, and where it is practically useful.

Simplistic proof extensions: The proofs are simple extensions from the linear regression case. I did not go through all the proofs, but the ones I did go through makes simple arguments that extrapolate the singular (non-multiple) instance case where each bag only has one instance.

**Q5 Detailed Comments To The Authors:**

There are several typos in the paper e.g. in the abstract itself: “which is matches”, please proof read. Since these typos do not compromise on the overall understanding of the paper and are easily fixable, I am not penalizing based on these. But if the authors could clarify the weaknesses it would be helpful .

**Q9 Complying With Reviewing Instructions:**

Yes

---

> ### Author Rebuttal · Authors · 2024-04-05
>
> We thank the Reviewer for their encouraging assessment of the paper's strengths and their helpful feedback. We address their concerns point-wise below.
>
> *Relevance and Practical Implications: There are almost no recent cited works (only 1 paper from last 3 years from the citations)*
> Authors: We list some recent applications of MIR across multiple areas. In a novel deployment of MIR, the work of (Serafini et al. 2022) used it to model electrical load disaggregation. In the biological domain, the work of (Park et al. 2020) uses MIR to model the continuous response of bags of neoantigens. For image quality assessment where each image patch has a probability of being prime, the work of (Liang at al. 2021) applied an MIR approach to train a CNN. A different image analysis task -  estimating facial age from images - has also been tackled using MIR techniques (Liu et al. 2019).
>
> We will add the above use-cases of MIR as well as the references below:
>
> (Serafini et al. 2022) L. Serafini, G. Tanoni, E. Principi, S. Spinsante and S. Squartini, "A Multiple Instance Regression Approach to Electrical Load Disaggregation", 2022 30th European Signal Processing Conference (EUSIPCO), 2022, pp. 1666-1670.
>
> (Park et al. 2020) Park S, Wang X, Lim J, Xiao G, Lu T, Wang T. “Bayesian multiple instance regression for modeling immunogenic neoantigens”. Statistical Methods in Medical Research. 2020;29(10):3032-3047.
>
> (Liang at al. 2021) Dong Liang, Xinbo Gao, Wen Lu, Jie Li, “Deep blind image quality assessment based on multiple instance regression”, Neurocomputing, Volume 431,2021,Pages 78-89.
>
> (Liu et al. 2019) Liu, J., Qiao, R., Li, Y., Li, S., “Witness detection in multi-instance regression and its application for age estimation”. Multimed Tools Appl 78, 33703–33722 (2019).
>
> $ $
>
> *Relevance and Practical Implications: the paper and the results are poorly motivated*
> Authors: We believe that MIR is of current practical relevance and any theoretical insights can impact the design and analysis of new techniques for MIR and are therefore interesting. Note that there are efficient algorithms to find the optimum linear regressor in the usual instance-wise (i.e., unit sized bags) case. In particular, given a set of point-label pairs which admit a perfect linear regressor one can easily find one (by solving a system of linear equations). Our hardness result (Theorems 1.2, 4.1) shows that this problem becomes hard in a strong sense when we have bags of size 2, and rules out a straightforward application of the simple techniques of the instance-wise case.
> Our generalization error bound (Theorems 1.1, 3.1) is the first such for the MIR problem - it shows that optimizing the bag-level MSE loss provably (in the case of random bags) learns the underlying instance labeling.
> This justifies our algorithmic approach --  Weighted Assignment model training -- of finding the prime instance assignment and the regressor to optimize the bag-level loss.  In the second to last paragraph before Section 1.1, we state our motivation for studying overlapping bags along with previously studied scenarios in which overlaps between bags arise. The above cited recent work (Serafini et al. 2022) also considers a novel scenario for MIR in which overlapping bags arise.
> We will appropriately add the above points early in the introduction so that the motivation and importance of our work is clear sooner in the paper's reading.
>
> $ $
>
> *Theory - Simplistic proof extensions*
> We respectfully disagree that the proof of the generalization error bound is a simplistic extension of the singular (unit-sized bags) case. Here, the main issue is that having bags affords more choice to a bad regressor $f$ - it can fit the bag-label by a low error prediction on any one of the instances in the bag. To show this is not possible with high probability for all bags, we show - using a bucketing argument - in the key Lemma 3.3 that among the $mk$ points sampled, there is a sizable subset $\mathcal{S}$ such that *all the values of $f$ on $\mathcal{S}$ are far from all the values of $f^\*$ on $\mathcal{S}$* ($\dagger$), where $f^\*$ is the instance-labeling.
> Lemma 3.4 shows that a significant fraction of the sampled bags are subsets of $\mathcal{S}$. These bags induce the lower bound on the bag-level loss since $f$ is bound to incur a high error on these bags due to the property ($\dagger$) above of $\mathcal{S}$.
> We will expand on the overview (Sec. 1.2)  including a more technical summary of the above.
> Also, the proof of Theorem 1.2 is fairly non-trivial, novel and yields a strong hardness of approximation result for MIR.
>
>
>
> $ $
>
> *Typos/Clarity*
> We thank the Reviewer for pointing out the typo, and sincerely regret these editorial issues. We assure the Reviewer that the paper will be proof read thoroughly and all the typos will be corrected in the final version.
>
> $ $
>
> We will open-source our experimental code along with the final version of the paper.

---

### Meta-Review · Area_Chair_ZUA5 · 2024-04-16

The focus of the submission is multiple instance regression (MIR; as phrased formally in Section 2.1), a regression problem where the inputs are bags of features in R^d, and the output label is determined by that of primary instances in the bags. The authors present two theoretical results (Theorem 3.1 and Theorem 4.1); the former is a bag-to-instance generalization bound, the latter is an NP-hardness result. They propose a weighted assignment based training technique which allows handling overlapping bags, and illustrate the effectiveness of the method on both synthetic and real-world examples (on the 1940 US Census Data).

MIR is a central problem of data science with numerous important applications, and the shown results are relevant, as it was assessed by the reviewers. The reviewers pointed out that at some parts the clarity/grammar of the submission could be improved, and putting the work better into context (elaboration of related work) would be beneficial.

Additional note connected to the last remark: There is also a large body of literature on a closely-related task called distribution regression, a regression problem where the inputs are probability distributions (specifically empirical measures corresponding to bags of features) with generalization guarantees. These methods date back to
- Barnabas Poczos, Alessandro Rinaldo, Aarti Singh, Larry Wasserman. Distribution-free distribution regression. In AISTATS, 31:507-515, 2013,
- Zoltan Szabo, Bharath Sriperumbudur, Barnabas Poczos, Arthur Gretton. Learning Theory for Distribution Regression. Journal of Machine Learning Research, 17(152):1-40, 2016.

I suggest elaborating the relation to distribution regression as well.